# MINDE: MUTUAL INFORMATION NEURAL DIFFUSION ESTIMATION

**Giulio Franzese**[1,*], **Mustapha Bounoua**[1,2], **Pietro Michiardi**[1]
[1]EURECOM,[2]Ampere Software Technology, *giulio.franzese@eurecom.fr

## ABSTRACT

In this work we present a new method for the estimation of Mutual Information (MI) between random variables. Our approach is based on an original interpretation of the Girsanov theorem, which allows us to use score-based diffusion models to estimate the Kullback-Leibler (KL) divergence between two densities as a difference between their score functions. As a by-product, our method also enables the estimation of the entropy of random variables. Armed with such building blocks, we present a general recipe to measure MI, which unfolds in two directions: one uses conditional diffusion process, whereas the other uses joint diffusion processes that allow simultaneous modelling of two random variables. Our results, which derive from a thorough experimental protocol over all the variants of our approach, indicate that our method is more accurate than the main alternatives from the literature, especially for challenging distributions. Furthermore, our methods pass MI self-consistency tests, including data processing and additivity under independence, which instead are a pain-point of existing methods. Code available.

## 1 INTRODUCTION

Mutual Information (MI) is a central measure to study the non-linear dependence between random variables [Shannon, 1948; MacKay, 2003], and has been extensively used in machine learning for representation learning [Bell & Sejnowski, 1995; Stratos, 2019; Belghazi et al., 2018; Oord et al., 2018; Hjelm et al., 2019], and for both training [Alemi et al., 2016; Chen et al., 2016; Zhao et al., 2018] and evaluating generative models [Alemi & Fischer, 2018; Huang et al., 2020].

For many problems of interest, precise computation of MI is not an easy task [McAllester & Stratos, 2020; Paninski, 2003], and a wide range of techniques for MI estimation have flourished. As the application of existing parametric and non-parametric methods [Pizer et al., 1987; Moon et al., 1995; Kraskov et al., 2004; Gao et al., 2015] to realistic, high-dimensional data is extremely challenging, if not unfeasible, recent research has focused on variational approaches [Barber & Agakov, 2004; Nguyen et al., 2007; Nowozin et al., 2016; Poole et al., 2019; Wunder et al., 2021; Letizia et al., 2023; Federici et al., 2023] and neural estimators [Papamakarios et al., 2017; Belghazi et al., 2018; Oord et al., 2018; Song & Ermon, 2019; Rhodes et al., 2020; Letizia & Tonello, 2022; Brekelmans et al., 2022] for MI estimation. In particular, the work by Song & Ermon [2019] and Federici et al. [2023] classify recent MI estimation methods into discriminative and generative approaches. The first class directly learns to estimate the ratio between joint and marginal densities, whereas the second estimates and approximates them separately.

In this work, we explore the problem of estimating MI using generative approaches, but with an original twist. In § 2 we review diffusion processes [Song et al., 2021] and in § 3 we explain how, thanks to the Girsanov Theorem [Øksendal, 2003], we can leverage score functions to compute the KL divergence between two distributions. This also enables the computation of the entropy of a random variable. In § 4 we present our general recipe for computing the MI between two arbitrary distributions, which we develop according to two modeling approaches, i.e., conditional and joint diffusion processes. The conditional approach is simple and capitalizes on standard diffusion models, but it is inherently more rigid, as it requires one distribution to be selected as the conditioning signal. Joint diffusion processes, on the other hand, are more flexible, but require an extension of traditional diffusion models, which deal with dynamics that allow data distributions to evolve according to multiple arrows of time.

Recent work by Czyż et al. [2023] argue that MI estimators are mostly evaluated assuming simple, multivariate normal distributions for which MI is analytically tractable, and propose a novel benchmark that introduces several challenges for estimators, such as sparsity of interactions, long-tailed distributions, invariance, and high mutual information. Furthermore, Song & Ermon [2019] introduce measures of self-consistency (additivity under independence and the data processing inequality) for MI estimators, to discern the properties of various approaches. In § 5 we evaluate several variants of our method, which we call Mutual Information Neural Diffusion Estimation (MINDE), according to such challenging benchmarks: our results show that MINDE outperforms the competitors on a majority of tasks, especially those involving challenging data distributions. Moreover, MINDE passes all self-consistency tests, a property that has remained elusive so far, for existing neural MI estimators.

## 2 DIFFUSION PROCESSES AND SCORE FUNCTIONS

We now revisit the theoretical background on diffusion processes, which is instrumental for the derivation of the methodologies proposed in this work. Consider the real space $\mathbb{R}^N$ and its associated Borel $\sigma-$algebra, defining the measurable space $\left(\mathbb{R}^N, \mathcal{B}(\mathbb{R}^N)\right)$. In this work, we consider Ito processes in $\mathbb{R}^N$ with duration $T < \infty$. Let $\Omega = D\left([0, T] \times \mathbb{R}^N\right)$, be the space of all $N-$dimensional continuous functions in the interval $[0, T]$, and the filtration $\mathcal{F}$ induced by the canonical process $X_t(\omega) = \omega_t, \omega \in \Omega$. As starting point, we consider an Ito process:

$$\begin{cases} \mathrm{d}X_t = f_t X_t \mathrm{d}t + g_t \mathrm{d}W_t, \\ X_0 = x \end{cases} \tag{1}$$

with given continuous functions $f_t \leq 0, g_t > 0$ and an arbitrary (deterministic) initial condition $x \in \mathbb{R}^N$. Equivalently, we can say that initial conditions are drawn from the Dirac measure $\delta_x$. This choice completely determines the path measure $\mathbb{P}^{\delta_x}$ of the corresponding probability space $\left(\Omega, \mathcal{F}, \mathbb{P}^{\delta_x}\right)$. Starting from $\mathbb{P}^{\delta_x}$ we construct a new path measure $\mathbb{P}^\mu$ by considering the product between $\mathbb{P}^{\delta_x}$ and measure $\mu$ in $\mathbb{R}^N$:

$$\mathbb{P}^\mu = \int_{\mathbb{R}^N} \mathbb{P}^{\delta_x} \mathrm{d}\mu(x). \tag{2}$$

Conversely, the original measure $\mathbb{P}^{\delta_x}$ can be recovered from $\mathbb{P}^\mu$ by conditioning the latter on the particular initial value $x$, i.e., the *projection* $\mathbb{P}^{\delta_x} = \mathbb{P}^\mu \#_x$. The new measure $\mathbb{P}^\mu$ can be represented by the following Stochastic Differential Equation (SDE):

$$\begin{cases} \mathrm{d}X_t = f_t X_t \mathrm{d}t + g_t \mathrm{d}W_t, \\ X_0 \sim \mu \end{cases} \tag{3}$$

associated to the corresponding probability spaces $(\Omega, \mathcal{F}, \mathbb{P}^\mu)$. We define $\nu_t^\mu$ as the pushforward of the complete path measure onto time instant $t \in [0, T]$, where by definition $\nu_0^\mu = \mu$.

It is instrumental for the scope of this work to study how the path measures and the SDEs representations change under *time reversal*. Let $\hat{X}_t \overset{\text{def}}{=} \omega_{T-t}$ be the time-reversed canonical process. If the canonical process $X_t$ is represented as in Eq. (3) under the path measure $\mathbb{P}^\mu$, then the time reversed process $\hat{X}_t$ has SDE representation [Anderson, 1982]:

$$\begin{cases} \mathrm{d}\hat{X}_t = -f_{T-t}\hat{X}_t + g_{T-t}^2 s_{T-t}^\mu(\hat{X}_t)\mathrm{d}t + g_{T-t}\mathrm{d}\hat{W}_t, \\ \hat{X}_0 \sim \nu_T^\mu \end{cases} \tag{4}$$

with corresponding path-reversed measure $\hat{\mathbb{P}}^\mu$, on the probability spaces with time-reversed filtration.

Next, we define the **score function** of the densities associated to the forward processes. In particular, $s_t^\mu(x) \overset{\text{def}}{=} \nabla \log\left(\bar{\nu}_t^\mu(x)\right)$, where $\bar{\nu}_t^\mu(x)$ is the density associated to the measures $\nu_t^\mu(x)$, computed with respect to the Lebesgue measure, $\mathrm{d}\nu_t^\mu(x) = \bar{\nu}_t^\mu(x)\mathrm{d}x$. In general we cannot assume exact knowledge of such *true* score function. Then, in practice, instead of the score function $s_t^\mu(x)$, we use *parametric* $(\theta)$ approximations thereof, $\tilde{s}_t^\mu(x)$, which we call the score network. Training the score network can be done by minimizing the following loss [Song et al., 2021; Huang et al., 2021; Kingma et al., 2021]:

$$\mathcal{L}(\theta) = \mathbb{E}_{\mathbb{P}^\mu}\left[\int_0^T \frac{g_t^2}{2}\left\|\tilde{s}_t^\mu(X_t) - \nabla \log\left(\bar{\nu}_t^{\delta_{X_0}}(X_t)\right)\right\|^2 \mathrm{d}t\right], \tag{5}$$

where $\nu_t^{\delta_{X_0}}$ stands for the measure of the processes at time $t$, conditioned on some initial value $X_0$.

## 3 KL DIVERGENCE AS DIFFERENCE OF SCORE FUNCTIONS

The MI between two random variables can be computed according to several equivalent expressions, which rely on the KL divergence between measures and/or entropy of measures. We then proceed to describe i) how to derive KL divergence between measures as the expected difference of score functions, ii) how to estimate such divergences given parametric approximation of the scores (and the corresponding estimation errors) and iii) how to cast the proposed methodology to the particular case of entropy estimation. In summary, this Section introduces the basic building blocks that we use in § 4 to define our MI estimators.

We consider the KL divergence between two generic measures $\mu^A$ and $\mu^B$ in $\mathbb{R}^N$, i.e. $\mathrm{KL}\left[\mu^A \parallel \mu^B\right]$, which is equal to $\int_{\mathbb{R}^N} \mathrm{d}\mu^A \log\left(\frac{\mathrm{d}\mu^A}{\mathrm{d}\mu^B}\right)$, if the Radon-Nikodym derivative $\frac{\mathrm{d}\mu^A}{\mathrm{d}\mu^B}$ exists (absolute continuity is satisfied), and $+\infty$ otherwise. Since our state space is $\mathbb{R}^N$, the following *disintegration* properties are valid [Léonard, 2014]:

$$\frac{\mathrm{d}\mathbb{P}^{\mu^A}}{\mathrm{d}\mathbb{P}^{\mu^B}}(\omega) = \frac{\mathrm{d}\left(\mathbb{P}^{\mu^A}\#\omega_0\right)}{\mathrm{d}\left(\mathbb{P}^{\mu^B}\#\omega_0\right)}(\omega)\frac{\mathrm{d}\mu^A(\omega_0)}{\mathrm{d}\mu^B(\omega_0)} = \frac{\mathrm{d}\mu^A(\omega_0)}{\mathrm{d}\mu^B(\omega_0)}, \frac{\mathrm{d}\hat{\mathbb{P}}^{\mu^A}}{\mathrm{d}\hat{\mathbb{P}}^{\mu^B}}(\omega) = \frac{\mathrm{d}\left(\hat{\mathbb{P}}^{\mu^A}\#\omega_T\right)}{\mathrm{d}\left(\hat{\mathbb{P}}^{\mu^B}\#\omega_T\right)}(\omega)\frac{\mathrm{d}\nu_T^{\mu^A}(\omega_T)}{\mathrm{d}\nu_T^{\mu^B}(\omega_T)},$$

$$(6)$$

where we implicitly introduced the product representation $\hat{\mathbb{P}}^{\mu^A} = \int_{\mathbb{R}^N} \hat{\mathbb{P}}_x \mathrm{d}\nu_T^{\mu^A}(x)$, similarly to Eq. (2).

Thanks to such disintegration theorems, we can write the KL divergence between the overall path measures $\mathbb{P}^{\mu^A}$ and $\mathbb{P}^{\mu^B}$ of two diffusion processes associated to the measures $\mu^A$ and $\mu^B$ as

$$\mathrm{KL}\left[\mathbb{P}^{\mu^A} \parallel \mathbb{P}^{\mu^B}\right] = \mathbb{E}_{\mathbb{P}^{\mu^A}}\left[\log\frac{\mathrm{d}\mathbb{P}^{\mu^A}}{\mathrm{d}\mathbb{P}^{\mu^B}}\right] = \mathbb{E}_{\mathbb{P}^{\mu^A}}\left[\log\frac{\mathrm{d}\mu^A}{\mathrm{d}\mu^B}\right] = \mathrm{KL}\left[\mu^A \parallel \mu^B\right], \quad (7)$$

where the second equality holds because, as observed on the left of Eq. (6), when conditioned on the same initial value, the path measures of the two forward processes coincide.

Now, since the KL divergence between the path measures is invariant to time reversal, i.e., $\mathrm{KL}\left[\mathbb{P}^{\mu^A} \parallel \mathbb{P}^{\mu^B}\right] = \mathrm{KL}\left[\hat{\mathbb{P}}^{\mu^A} \parallel \hat{\mathbb{P}}^{\mu^B}\right]$, using similar disintegration arguments, it holds that:

$$\mathrm{KL}\left[\hat{\mathbb{P}}^{\mu^A} \parallel \hat{\mathbb{P}}^{\mu^B}\right] = \mathbb{E}_{\hat{\mathbb{P}}^{\mu^A}}\left[\log\frac{\mathrm{d}\left(\hat{\mathbb{P}}^{\mu^A}\#\omega_T\right)}{\mathrm{d}\left(\hat{\mathbb{P}}^{\mu^B}\#\omega_T\right)}\right] + \mathbb{E}_{\hat{\mathbb{P}}^{\mu^A}}\left[\log\frac{\mathrm{d}\nu_T^{\mu^A}}{\mathrm{d}\nu_T^{\mu^B}}\right]. \quad (8)$$

The first term on the r.h.s of Eq. (8) can be computed using the Girsanov theorem [Øksendal, 2003] as

$$\mathbb{E}_{\hat{\mathbb{P}}^{\mu^A}}\left[\int_0^T \frac{1}{2g_t^2}\left\|g_t^2\left(s_t^{\mu^A}(\hat{X}_t) - s_t^{\mu^B}(\hat{X}_t)\right)\right\|^2 \mathrm{d}t\right] = \mathbb{E}_{\mathbb{P}^{\mu^A}}\left[\int_0^T \frac{g_t^2}{2}\left\|s_t^{\mu^A}(X_t) - s_t^{\mu^B}(X_t)\right\|^2 \mathrm{d}t\right]. \quad (9)$$

The second term on the r.h.s of Eq. (8), equals $\mathrm{KL}\left[\nu_T^{\mu^A} \parallel \nu_T^{\mu^B}\right]$: this is a vanishing term with $T$, i.e. $\lim_{T\to\infty} \mathrm{KL}\left[\nu_T^{\mu^A} \parallel \nu_T^{\mu^B}\right] = 0$. To ground this claim, we borrow the results by Collet & Malrieu [2008], which hold for several forward diffusion SDEs of interest, such as the Variance Preserving (VP), or Variance Exploding (VE) SDEs Song et al. [2021]. In summary, it is necessary to adapt the classical Bakry-Émery condition of diffusion semigroup to the non homogeneous case, and exploit the contraction properties of diffusion on the KL divergences.

Combining the different results, we have that:

$$\mathrm{KL}\left[\mu^A \parallel \mu^B\right] = \mathbb{E}_{\mathbb{P}^{\mu^A}}\left[\int_0^T \frac{g_t^2}{2}\left\|s_t^{\mu^A}(X_t) - s_t^{\mu^B}(X_t)\right\|^2 \mathrm{d}t\right] + \mathrm{KL}\left[\nu_T^{\mu^A} \parallel \nu_T^{\mu^B}\right] \quad (10)$$

which constitutes the basic equality over which we construct our estimators, described in § 3.1.

We conclude by commenting on the possibility of computing divergences in a *latent* space. Indeed, in many natural cases, the density $\mu^A$ is supported on a lower dimensional manifold $\mathcal{M} \subset \mathbb{R}^N$ [Loaiza-Ganem et al., 2022]. Whenever we can find encoder and decoder functions $\psi, \phi$, respectively, such that $\phi(\psi(x)) = x, \mu^A$ — almost surely, and $\phi(\psi(x)) = x, \mu^B$ — almost surely, the KL divergence can be computed in the *latent* space obtained by the encoder $\psi$. Considering the pushforward measure $\mu^A \circ \psi^{-1}$, it is indeed possible to show (proof in § A) that $\mathrm{KL}\left[\mu^A \parallel \mu^B\right] = \mathrm{KL}\left[\mu^A \circ \psi^{-1} \parallel \mu^B \circ \psi^{-1}\right]$. This property is particularly useful as it allows using score based models trained in a latent space to compute the KL divergences of interest, as we do in § 5.2.

## 3.1 KL ESTIMATORS AND THEORETICAL GUARANTEES

Given the parametric approximations of the score networks through minimization of Eq. (5), and the result in Eq. (10), we are ready to discuss our proposed **estimator** of the KL divergence. We focus on the first term on the r.h.s. of Eq. (10), which has unknown value, and define its approximated version

$$
e(\mu^A, \mu^B) \overset{\text{def}}{=} \mathbb{E}_{\mathbb{P}_{\mu^A}}\left[\int_0^T \frac{g_t^2}{2}\left\|\tilde{s}_t^{\mu^A}(X_t) - \tilde{s}_t^{\mu^B}(X_t)\right\|^2 \mathrm{d}t\right] = \int_0^T \frac{g_t^2}{2}\mathbb{E}_{\nu_t^{\mu^A}}\left[\left\|\tilde{s}_t^{\mu^A}(X_t) - \tilde{s}_t^{\mu^B}(X_t)\right\|^2\right]\mathrm{d}t,
$$
(11)

where parametric scores, instead of true score functions, are used. By defining the score error as $\epsilon_t^{\mu^A}(x) \overset{\text{def}}{=} \tilde{s}_t^{\mu^A}(x) - s_t^{\mu^A}(x)$, it is possible to show (see § A) that $e(\mu^A, \mu^B) - \mathbb{E}_{\mathbb{P}_{\mu^A}}\left[\int_0^T \frac{g_t^2}{2}\left\|s_t^{\mu^A}(X_t) - s_t^{\mu^B}(X_t)\right\|^2 \mathrm{d}t\right]$ has expression

$$
d = \mathbb{E}_{\mathbb{P}_{\mu^A}}\left[\int_0^T \frac{g_t^2}{2}\left\|\epsilon_t^{\mu^A}(X_t) - \epsilon_t^{\mu^B}(X_t)\right\|^2 + 2\langle s_t^{\mu^A}(X_t) - s_t^{\mu^B}(X_t), \epsilon_t^{\mu^A}(X_t) - \epsilon_t^{\mu^B}(X_t)\rangle \mathrm{d}t\right].
$$
(12)

As for the second term on the r.h.s. of Eq. (10), $\mathrm{KL}\left[\nu_T^{\mu^A} \parallel \nu_T^{\mu^B}\right]$, we recall that it is a quantity that vanishes with large $T$. Consequently, given a sufficiently large diffusion time $T$ the function $e$ serves as an accurate estimator of the true KL:

$$
e(\mu^A, \mu^B) = \mathrm{KL}\left[\mu^A \parallel \mu^B\right] + d - \mathrm{KL}\left[\nu_T^{\mu^A} \parallel \nu_T^{\mu^B}\right] \simeq \mathrm{KL}\left[\mu^A \parallel \mu^B\right].
$$
(13)

An important property of our estimator is that it is *neither* an upper nor a lower bound of the true KL divergence: indeed the $d$ term of Eq. (13) can be either positive or negative. This property, frees our estimation guarantees from the pessimistic results of McAllester & Stratos [2020]. Note also that, counter-intuitively, larger errors norms $\left\|\epsilon_t^{\mu^A}(x)\right\|$ not necessarily imply larger estimation error of the KL divergence. Indeed, common mode errors (reminiscent of paired statistical tests) cancel out. In the special case where $\epsilon_t^{\mu^A}(x) = \epsilon_t^{\mu^B}(x)$, the estimation error due to the approximate nature of the score functions is indeed zero.

Accurate quantification of the estimation error is, in general, a challenging task. Indeed, techniques akin to the works [De Bortoli, 2022; Lee et al., 2022; Chen et al., 2022], where guarantees are provided w.r.t. to the distance between the real backward dynamics and the measures induced by the *simulated* backward dynamics, $\mathrm{KL}\left[\mu^A \parallel \tilde{\mu}^A\right]$, are not readily available in our context. Qualitatively, we observe that our estimator is affected by two sources of error: score networks that only approximate the true score function and finiteness of $T$. The $d$ term in Eq. (13), which is related to the score discrepancy, suggests selection of a small time $T$ (indeed we can expect such mismatch to behave as a quantity that increases with $T$ [Franzese et al., 2023]). It is important however to adopt a sufficiently large diffusion time $T$ such that $\mathrm{KL}\left[\nu_T^{\mu^A} \parallel \nu_T^{\mu^B}\right]$ is negligible. Typical diffusion schedules satisfy these requirements. Note that, if the KL term is known (or approximately known), it can be included in the definition of the estimator function, reducing the estimation error (see also discussion in § 3.2).

**Montecarlo Integration** The analytical computation of Eq. (11) is, in general, out of reach. However, Montecarlo integration is possible, by recognizing that samples from $\nu_t^{\mu^A}$ can be obtained through the sampling scheme $X_0 \sim \mu^A$, $X_t \sim \nu_t^{\delta_{X_0}}$. The outer integration w.r.t. to the time instant is similarly possible by sampling $t \sim \mathcal{U}(0,T)$, and multiplying the result of the estimation by $T$ (since $\int_0^T (\cdot) \mathrm{d}t = T \mathbb{E}_{t \sim \mathcal{U}(0,T)}[(\cdot)]$). Alternatively, it is possible to implement importance sampling schemes to reduce the variance, along the lines of what described by Huang et al. [2021], by sampling the time instant non-uniformly and modifying accordingly the time-varying constants in Eq. (11). In both cases, the Montecarlo estimation error can be reduced to arbitrary small values by collecting enough samples, with guarantees described in [Rainforth et al., 2018].

## 3.2 ENTROPY ESTIMATION

We now describe how to compute the entropy associated to a given density $\mu^A$, $\mathrm{H}(\mu^A) \stackrel{\text{def}}{=} \int \mathrm{d}\mu^A(x) \log \bar{\mu}^A(x)$. Using the ideas for estimating the KL divergence, we notice that we can compute $\mathrm{KL}\left[\mu^A \parallel \gamma_\sigma\right]$, where $\bar{\gamma}_\sigma(x)$ stands for the standard Gaussian distribution with mean $0$ and covariance $\sigma^2 I$. Then, we can relate the entropy to such divergence through the following equality:

$$\mathrm{H}(\mu^A) + \mathrm{KL}\left[\mu^A \parallel \gamma_\sigma\right] = -\int \mathrm{d}\mu^A(x) \log \bar{\gamma}_\sigma(x) = \frac{N}{2} \log(2\pi\sigma^2) + \frac{\mathbb{E}_{\mu^A}\left[X_0^2\right]}{2\sigma^2}. \quad (14)$$

A simple manipulation of Eq. (14), using the results from § 3.1, implies that the estimation of the entropy $\mathrm{H}(\mu^A)$ involves three unknown terms: $e(\mu^A, \gamma_\sigma)$, $\mathrm{KL}\left[\nu_T^{\mu^A} \parallel \nu_T^{\gamma_\sigma}\right]$ and $\frac{\mathbb{E}_{\mu^A}\left[X_0^2\right]}{2\sigma^2}$. Now, the score function associated to the forward process starting from $\gamma_\sigma$ is analytically known and has value $s_t^{\gamma_\sigma}(x) = -\chi_t^{-1}x$, where $\chi_t = \left(k_t^2\sigma^2 + k_t^2 \int_0^t k_s^{-2} g_s^2 \mathrm{d}s\right) I$, with $k_t = \exp\left\{\left(\int_0^t f_s \mathrm{d}s\right)\right\}$. Moreover, whenever $T$ is large enough $\nu_T^{\mu^A} \simeq \gamma_1$, independently on the chosen value of $\sigma$. Consequently $\mathrm{KL}\left[\nu_T^{\mu^A} \parallel \nu_T^{\gamma_\sigma}\right] \simeq \mathrm{KL}\left[\gamma_1 \parallel \gamma_{\sqrt{\chi_T}}\right]$, which is analytically available as $N/2\left(\log(\chi_T) - 1 + 1/\chi_T\right)$. Quantification of such approximation is possible following the same lines defined by Collet & Malrieu [2008]. In summary, we consider the following estimator for the entropy:

$$\mathrm{H}(\mu^A; \sigma) \simeq \frac{N}{2} \log(2\pi\sigma^2) + \frac{\mathbb{E}_{\mu^A}\left[X_0^2\right]}{2\sigma^2} - e(\mu^A, \gamma_\sigma) - \frac{N}{2}\left(\log(\chi_T) - 1 + \frac{1}{\chi_T}\right) \quad (15)$$

For completeness, we note that a related estimator has recently appeared in the literature [Kong et al., 2022], although the technical derivation and objectives are different than ours.

## 4 COMPUTATION OF MUTUAL INFORMATION

In this work, we are interested in estimating the MI between two random variables $A, B$. Consequently, we need to define the joint, conditional and marginal measures. We consider the first random variable $A$ in $\mathbb{R}^N$ to have marginal measure $\mu^A$. Similarly, we indicate the marginal measure of the second random variable $B$ with $\mu^B$. The joint measure of the two random variables $C \stackrel{\text{def}}{=} [A, B]$, which is defined in $\mathbb{R}^{2N}$, is indicated with $\mu^C$. What remains to be specified are the conditional measures of the first variable given a particular value of the second $A \mid B = y$, shortened with $A_y$, that we indicate with the measure $\mu^{A_y}$, and the conditional measure of the second given a particular value of the first, $B \mid A = x$, shortened with $B_x$, and indicated with $\mu^{B_x}$. This choice of notation, along with Bayes theorem, implies the following set of equivalences: $\mathrm{d}\mu^C(x, y) = \mathrm{d}\mu^{A_y}(x)\mathrm{d}\mu^B(y) = \mathrm{d}\mu^{B_x}(y)\mathrm{d}\mu^A(x)$ and $\mu^A = \int \mu^{A_y}\mathrm{d}\mu^B(y)$, $\mu^B = \int \mu^{B_x}\mathrm{d}\mu^A(x)$.

The marginal measures $\mu^A, \mu^B$ are associated to diffusion of the form of Eq. (3). Similarly, the joint $\mu^C$ and conditional $\mu^{A_y}$ measures we introduced, are associated to forward diffusion processes:

$$\begin{cases} \mathrm{d}[X_t, Y_t]^\top = f_t[X_t, Y_t]^\top \mathrm{d}t + g_t[\mathrm{d}W_t, \mathrm{d}W_t']^\top \\ [X_0, Y_0]^\top \sim \mu^C \end{cases} , \quad \begin{cases} \mathrm{d}X_t = f_t X_t \mathrm{d}t + g_t \mathrm{d}W_t \\ X_0 \sim \mu^{A_y} \end{cases} \quad (16)$$

respectively, where the SDE on the l.h.s. is valid for the real space $\mathbb{R}^{2N}$, as defined in § 2.

In this work, we consider two classes of diffusion processes. In the first case, the diffusion model is asymmetric, and the random variable $B$ is only considered as a conditioning signal. As such, we learn the score associated to the random variable $A$, with a conditioning signal $B$, which is set to some predefined null value when considering the marginal measure. This well-known approach [Ho & Salimans, 2021] effectively models the marginal and conditional scores associated to $\mu^A$ and $\mu^{A_y}$ with a unique score network.

Next, we define a new kind of diffusion model for the joint random variable $C$, which allows modelling the joint and the conditional measures. Inspired by recent trends in multi-modal generative modeling [Bao et al., 2023; Bounoua et al., 2023], we define a joint diffusion process that allows amortized training of a single score network, instead of considering separate diffusion processes and their respective score networks, for each random variable. To do so, we define the following SDE:

$$\begin{cases} \mathrm{d}\,[X_t, Y_t]^\top = f_t[\alpha X_t, \beta Y_t]^\top \mathrm{d}t + g_t\,[\alpha \mathrm{d}W_t, \beta \mathrm{d}W_t']^\top\,, \\ [X_0, Y_0]^\top \sim \mu^C, \end{cases} \tag{17}$$

with extra parameters $\alpha, \beta \in \{0, 1\}$. This SDE extends the l.h.s. of Eq. (16), and describes the joint evolution of the variables $X_t, Y_t$, starting from the joint measure $\mu^C$, with overall path measure $\mathbb{P}^{\mu^C}$. The two extra coefficients $\alpha, \beta$ are used to modulate the *speed* at which the two portions $X_t, Y_t$ of the process diffuse towards their steady state. More precisely, $\alpha = \beta = 1$ corresponds to a *classical* simultaneous diffusion (l.h.s. of Eq. (16)). On the other hand, the configuration $\alpha = 1, \beta = 0$ corresponds to the case in which the variable $Y_t$ remains constant throughout all the diffusion (which is used for conditional measures, r.h.s. of Eq. (16)). The specular case, $\alpha = 0, \beta = 1$, similarly allows to study the evolution of $Y_t$ conditioned on a constant value of $X_0$. Then, instead of learning three separate score networks (for $\mu^C, \mu^{A_y}$ and $\mu^{B_x}$), associated to standard diffusion processes, the key idea is to consider a *unique* parametric score, leveraging the unified formulation Eq. (17), which accepts as inputs two vectors in $\mathbb{R}^N$, the diffusion time $t$, and the two coefficients $\alpha, \beta$. This allows to conflate in a single architecture: i) the score $s_t^{\mu^C}(x, y)$ associated to the joint diffusion of the variables $A, B$ (corresponding to $\alpha = \beta = 1$) and ii) the conditional score $s_t^{\mu^{A_y}}(x)$ (corresponding to $\alpha = 1, \beta = 0$). Additional details are presented in § C.

### 4.1 MINDE: A FAMILY OF MI ESTIMATORS

We are now ready to describe our new MI estimator, which we call MINDE. As a starting point, we recognize that the MI between two random variables $A, B$ has several equivalent expressions, among which Eqs. (18) to (20). On the left hand side of these expressions we report well-known formulations for the MI, $\mathrm{I}(A, B)$, while on the right hand side we express them using the estimators we introduce in this work, where equality is assumed to be valid up to the errors described in § 3.

$$\mathrm{H}(A) - \mathrm{H}(A \mid B) \simeq - e(\mu^A, \gamma_\sigma) + \int e(\mu^{A_y}, \gamma_\sigma)\mathrm{d}\mu^B(y), \tag{18}$$

$$\int \mathrm{KL}\left[\mu^{A_y} \,\|\, \mu^A\right] \mathrm{d}\mu^B(y) \simeq \int e(\mu^{A_y}, \mu^A)\mathrm{d}\mu^B(y), \tag{19}$$

$$\mathrm{H}(C) - \mathrm{H}(A \mid B) - \mathrm{H}(B \mid A) \simeq - e(\mu^C, \gamma_\sigma) + \int e(\mu^{A_y}, \gamma_\sigma)\mathrm{d}\mu^B(y) + \int e(\mu^{B_x}, \gamma_\sigma)\mathrm{d}\mu^A(x). \tag{20}$$

Note that it is possible to derive (details in § B) another equality for the MI:

$$\mathrm{I}(A, B) \simeq \mathbb{E}_{\mathbb{P}^{\mu^C}}\left[\int_0^T \frac{g_t^2}{2}\left[\left\|\tilde{s}_t^{\mu^C}([X_t, Y_t]) - [\tilde{s}_t^{\mu^{A_{Y_0}}}(X_t), \tilde{s}_t^{\mu^{B_{X_0}}}(Y_t)]\right\|^2\right]\mathrm{d}t\right]. \tag{21}$$

Next, we describe how the conditional and joint modeling approaches can be leveraged to compute a *family* of techniques to estimate MI. We evaluate all the variants in § 5.

**Conditional Diffusion Models.** We start by considering conditional models. A simple MI estimator can be obtained considering Eq. (18). The entropy of $A$ can be estimated using Eq. (15). Similarly,

we can estimate the conditional entropy $H(A \mid B)$ using the equality $H(A \mid B) = \int H(A_y) \mathrm{d} \mu^B(y)$, where the argument of the integral, $H(A_y)$, can be again obtained using Eq. (15). Notice, that since $\mathbb{E}_{\mu^B(y)} \mathbb{E}_{\mu^{A_y}} \left[ X_0^2 \right] = \mathbb{E}_{\mu^A} \left[ X_0^2 \right]$, when substracting the estimators of $H(A)$ and $H(A \mid B)$, all the terms but the estimator functions $e(\cdot)$ cancels out, leading to the equality in Eq. (18). A second option is to simply use Eq. (19) and leverage Eq. (11).

**Joint diffusion models.** Armed with the definition of a joint diffusion processes, and the corresponding score function, we now describe the basic ingredients that allow estimation of the MI, according to various formulations. Using the joint score function $s_t^{\mu^C}([x, y])$, the estimation of the joint entropy $H(A, B)$ can be obtained with a straightforward application of Eq. (15). Similarly, the conditional entropy $H(A \mid B) = \int H(A_y) \mathrm{d} \mu^B(y)$ can be computed using $s_t^{\mu^{A_y}}(x)$ to obtain the conditional score. Notice that $H(B \mid A)$ is similarly obtained. Given the above formulations of the joint and conditional entropy, it is now easy to compute the MI according to Eq. (20), where we notice that, similarly to what discussed for conditional models, many of the terms in the different entropy estimations cancel out. Finally, it is possible to compute the MI according to Eq. (21). Interestingly, this formulation allows to eliminate the need for the parameter $\sigma$ of the entropy estimators, similarly to the MINDE conditional variant, which shares this property as well (Eq. (18)).

## 5 EXPERIMENTAL VALIDATION

We now evaluate the different estimators proposed in § 4. In particular, we study conditional and joint models (MINDE-C and MINDE-J respectively), and variants that exploit the difference between the parametric scores *inside* the same norm ( Eqs. (19) and (21)) or *outside* it, adopting the difference of entropies representation along with Gaussian reference scores $s^{\gamma_c}$ (Eqs. (18) and (20)). Summarizing, we refer to the different variants as MINDE-C($\sigma$), MINDE-C, and MINDE-J($\sigma$), MINDE-J, for Eqs. (18) to (21) respectively. Our empirical validation involves a large range of synthetic distributions, which we present in § 5.1. We also analyze the behavior of all MINDE variants according to *self-consistency* tests, as discussed in § 5.2.

For all the settings, we use a simple, stacked multi-layer perception (MLP) with skip connections adapted to the input dimensions, and adopt VP-SDE diffusion Song et al. [2021]. We apply importance sampling [Huang et al., 2021; Song et al., 2021] at both training and inference time. More details about the implementation are included in § C.

### 5.1 MI ESTIMATION BENCHMARK

We use the evaluation strategy proposed by Czyż et al. [2023], which covers a range of distributions going beyond what is typically used to benchmark MI estimators, e.g., multivariate normal distributions. In summary, we consider high-dimensional cases with (possibly) long-tailed distributions and/or sparse interactions, in the presence of several non trivial non-linear transformation. Benchmarks are constructed using samples from several base distributions, including Uniform, Normal with either dense or sparse correlation structure, and long-tailed Student distributions. Such samples are further modified by deterministic transformations, including the Half-Cube homeomorphism, which extends the distribution tails, and the Asinh Mapping, which instead shortens them, the Swiss Roll Embedding and Spiral diffeomorphis, which alter the simple linear structure of the base distributions.

We compare MINDE against neural estimators, such as MINE [Belghazi et al., 2018], INFONCE [Oord et al., 2018], NWJ [Nguyen et al., 2007] and DOE [McAllester & Stratos, 2020]. To ensure a fair comparison between MINDE and other neural competitors, we consider architectures with a comparable number of parameters. Note that the original benchmark in [Czyż et al., 2023] uses 10k training samples, which are in many cases not sufficient to obtain stable estimates of the MI for our competitors. Here, we use a larger training size (100k samples) to avoid confounding factors in our analysis. In all our experiments, we fix $\sigma = 1.0$ for the MINDE-C($\sigma$), MINDE-J($\sigma$) variants, which results in the best performance (an ablation study is included in § D).

**Results:** The general benchmark consists of 40 tasks (10 unique tasks $\times$ 4 parametrizations) designed by combining distributions and MI-invariant transformations discussed earlier. We average results over 10 seeds for MINDE variants and competitors, following the same protocol as in Czyż

et al. [2023]. We present the full set of MI estimation tasks in Table 1. As in the original Czyż et al. [2023], estimates for the different methods are presented with a precision of 0.1 nats, to improve visualization. For low-dimensional distributions, benchmark results show that all methods are effective in accurate MI estimation. Differences emerge for more challenging scenarios. Overall, all our MINDE variants perform well. MINDE–C stands out as the best estimator with 35/40 estimated tasks with an error within the 0.1 nats quantization range. Moreover, MINDE can accurately estimate the MI for long tailed distributions (Student) and highly transformed distributions (Spiral, Normal CDF), which are instead problematic for most of the other methods. The MINE estimator achieves the second best performance, with an MI estimation within 0.1 nats from ground truth for 24/40 tasks. Similarly to the other neural estimator baselines, MINE is limited when dealing with long tail distributions (Student), and significantly transformed distributions (Spiral).

**High MI benchmark:** Through this second benchmark, we target high MI distributions. We consider $3 \times 3$ multivariate normal distribution with sparse interactions as done in Czyż et al. [2023]. We vary the correlation parameter to obtain the desired MI, and test the estimators when applying Half-cube or Spiral transformations. Results in Figure 1 show that while on the non transformed distribution (column (a)) all neural estimators nicely follow the ground truth, on the transformed versions (columns (b) and (c)), MINDE outperforms competitors.

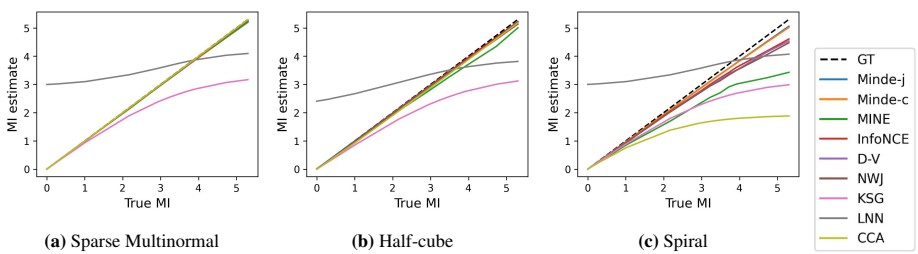

| (a) Sparse Multinormal | (b) Half-cube | (c) Spiral |

**Figure 1:** High MI benchmark: original (column (a)) and transformed variants (columns (b) and (c)).

## 5.2 Consistency tests

The second set of tests we perform are the self-consistency ones proposed in Song & Ermon [2019], which aim at investigating properties of MI estimators on real data. Considering as random variable $A$ a sample from the MNIST (resolution $28 \times 28$) dataset, the first set of measurements performed is the estimation of $I(A, B_r)$, where $B_r$ is equal to $A$ for the first $r$ rows, and set to 0 afterwards. It is evident that $I(A, B_r)$ is a quantity that increases with $r$, where in particular $I(A, B_0) = 0$. Testing whether this holds also for the estimated MI is referred to as *independency* test. The second test proposed in Song & Ermon [2019] is the *data-processing* test, where given that $I(A; [B_{r+k}, B_r]) = I(A; B_{r+k}), k > 0$, the task is to verify it through estimators for different values of $k$. Finally, the *additivity* tests aim at assessing whether for two independent images $A^1, A^2$ extracted from the dataset, the property $I([A^1, A^2]; [B_r^1, B_r^2]) = 2I(A^1; B_r^1)$ is satisfied also by the numerical estimations.

For these tests, we consider diffusion models in a latent space, exploiting the invariance of KL divergences to perfect auto-encoding (see § 3). First, we train for all tests deterministic auto-encoders for the considered images. Then, through concatenation of the latent variables, as done in [Bao et al., 2023; Bounoua et al., 2023], we compute the MI with the different schemes proposed in § 4. Results of the three tests (averaged over 5 seeds) are reported in Figure 2. In general, all MINDE variants show excellent performance, whereas none of the other neural MI estimators succeed at passing simultaneously all tests, as can be observed from Figures 4,5,6 in the original Song & Ermon [2019]).

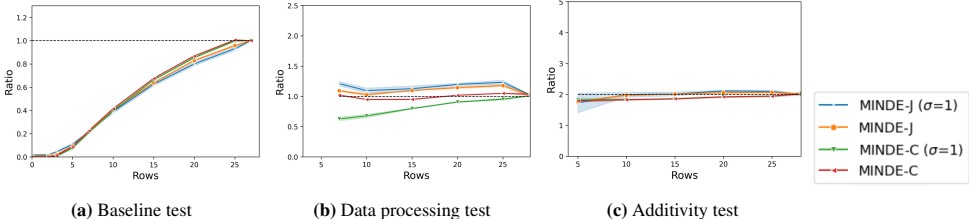

| (a) Baseline test | (b) Data processing test | (c) Additivity test |

**Figure 2:** Consistency tests results on the MNIST dataset. *Baseline test Figure 2a:* Evaluation of $\frac{I(A,B_r)}{I(A,B_0)}$. $A$ is an image and $B_r$ is an image containing the top $t$ rows of $A$. *Data processing test Figure 2b:* Evaluation of $\frac{I(A,[B_{r+k},B_r])}{I(A,B_{r+k})}$ (ideal value is 1). *Additivity test Figure 2c:* Evaluation of $\frac{I([A^1,A^2],[B_r^1,B_r^2])}{I(A^1,B_r^1)}$ (ideal value is 2).

**Table 1:** Mean MI estimates over 10 seeds using N = 10k test samples against ground truth (GT). Color indicates relative negative (red) and positive bias (blue). All methods were trained with 100k samples. List of abbreviations ( *Mn*: Multinormal, *St*: Student-t, *Nm*: Normal, *Hc*: Half-cube, *Sp*: Spiral)

## 6 CONCLUSION

The estimation of MI stands as a fundamental goal in many areas of machine learning, as it enables understanding the relationships within data, driving representation learning, and evaluating generative models. Over the years, various methodologies have emerged to tackle the difficult task of MI estimation, addressing challenges posed by high-dimensional, real-world data. Our work introduced a novel method, MINDE, which provides a unique perspective on MI estimation by leveraging the theory of diffusion-based generative models. We expanded the classical toolkit for information-theoretic analysis, and showed how to compute the KL divergence and entropy of random variables using the score of data distributions. We defined several variants of MINDE, which we have extensively tested according to a recent, comprehensive benchmark that simulates real-world challenges, including sparsity, long-tailed distributions, invariance to transformations. Our results indicated that our methods outperform state-of-the-art alternatives, especially on the most challenging tests. Additionally, MINDE variants successfully passed self-consistency tests, validating the robustness and reliability of our proposed methodology.

Our research opens up exciting avenues for future exploration. One compelling direction is the application of MINDE to large-scale multi-modal datasets. The conditional version of our approach enables harnessing the extensive repository of existing pre-trained diffusion models. For instance, it could find valuable application in the estimation of MI for text-conditional image generation. Conversely, our joint modeling approach offers a straightforward path to scaling MI estimation to more than two variables. A scalable approach to MI estimation is particularly valuable when dealing with complex systems involving multiple interacting variables, eliminating the need to specify a hierarchy among them.

## ACKNOWLEDGEMENTS

GF and PM gratefully acknowledges support from Huawei Paris and the European Commission (ADROIT6G Grant agreement ID: 101095363).

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

MINDE: Mutual Information Neural Diffusion Estimation —
Supplementary material

## A    Proofs of § 3

*Proof of Auto-encoder invariance of* KL. Whenever we can find encoder and decoder functions $\phi, \psi$ respectively such that $\phi(\psi(x)) = x, \mu^A-$ almost surely and $\phi(\psi(x)) = x, \mu^B-$ almost surely, the Kullback-Leibler divergence can be computed in the *latent* space obtained by the encoder $\psi$:

$$\text{KL}\left[\mu^A \parallel \mu^B\right] = \int_{\mathcal{M}} \log \frac{d\mu^A}{d\mu^B} d\mu^A =$$

$$\int_{\mathcal{M}} \log \left(\frac{d\mu^A}{d\mu^B} \circ \phi \circ \psi\right) d\mu^A = \int_{\psi(\mathcal{M})} \log \left(\frac{d\mu^A}{d\mu^B} \circ \phi\right) d\left(\mu^A \circ \psi^{-1}\right) =$$

$$\int_{\psi(\mathcal{M})} \log \left(\frac{d\mu^A}{d\mu^B} \circ \psi^{(-1)}\right) d\left(\mu^A \circ \psi^{-1}\right) = \text{KL}\left[\tilde{\mu}^A \parallel \tilde{\mu}^B\right]. \tag{22}$$

*Proof of Eq.* (12). To prove such claim, it is sufficient to start from the r.h.s. of Eq. (11), substitute to the parametric scores their definition with the errors $\epsilon_t^{\mu^A}(x) = \tilde{s}_t^{\mu^A}(x) - s_t^{\mu^A}(x)$, and expand the square:

$$\int_0^T \frac{g_t^2}{2} \mathbb{E}_{\nu_t^{\mu^A}} \left[\left\|\tilde{s}_t^{\mu^A}(X_t) - \tilde{s}_t^{\mu^B}(X_t)\right\|^2\right] dt =$$

$$\int_0^T \frac{g_t^2}{2} \mathbb{E}_{\nu_t^{\mu^A}} \left[\left\|s_t^{\mu^A}(X_t) + \epsilon_t^{\mu^A}(x) - s_t^{\mu^B}(X_t) - \epsilon_t^{\mu^B}(x)\right\|^2\right] dt =$$

$$\int_0^T \frac{g_t^2}{2} \mathbb{E}_{\nu_t^{\mu^A}} \left[\left\|s_t^{\mu^A}(X_t) - s_t^{\mu^B}(X_t)\right\|^2\right] dt +$$

$$\int_0^T \frac{g_t^2}{2} \mathbb{E}_{\nu_t^{\mu^A}} \left[\left\|\epsilon_t^{\mu^A}(X_t) - \epsilon_t^{\mu^B}(X_t)\right\|^2 + 2\langle s_t^{\mu^A}(X_t) - s_t^{\mu^B}(X_t) +, \epsilon_t^{\mu^A}(X_t) - \epsilon_t^{\mu^B}(X_t)\rangle\right] dt,$$

from which the definition of $d$ holds.

## B    Proof of Eq. (21)

We start with the approximation of Eq. (20):

$$I(A, B) \simeq -e(\mu^C, \gamma_\sigma) + \int e(\mu^{A_y}, \gamma_\sigma) d\mu^B(y) + \int e(\mu^{B_x}, \gamma_\sigma) d\mu^A(x). \tag{23}$$

Since the approximation is valid for any $\sigma$, we select the limit of $\sigma \to \infty$, where the reference score $\chi_t^{-1} x$ converges to zero, and can thus be neglected from the estimators integral (for example,

$e(\mu^A, \gamma_\infty) \simeq \int\limits_0^T \frac{g_t^2}{2} \mathbb{E}_{\nu_t^{\mu^A}}\left[\left\|\tilde{s}_t^{\mu^A}(X_t)\right\|^2\right] \mathrm{d}t)$. This allows to obtain:

$$\mathrm{I}(A, B) \simeq -\int\limits_0^T \frac{g_t^2}{2} \int \mathrm{d}\nu_t^{\mu^C}([x,y])\left\|\tilde{s}_t^{\mu^C}([x,y])\right\|^2 \mathrm{d}t+$$

$$\int \left(\int\limits_0^T \frac{g_t^2}{2} \int \mathrm{d}\nu_t^{\mu^{Ay}}(x)\left\|\tilde{s}_t^{\mu^{Ay}}(x)\right\|^2 \mathrm{d}t\right) \mathrm{d}\mu^B(y)+$$

$$\int \left(\int\limits_0^T \frac{g_t^2}{2} \int \mathrm{d}\nu_t^{\mu^{Bx}}(y)\left\|\tilde{s}_t^{\mu^{Bx}}(y)\right\|^2 \mathrm{d}t\right) \mathrm{d}\mu^A(x).$$

As a further step in the derivation of our approximation, we consider the estimated scores to be sufficiently good, such that we substitute the parametric with the true scores. In such case, the following holds:

$\mathrm{I}(A, B) \simeq$

$$\int\limits_0^T \frac{g_t^2}{2} \int \mathrm{d}\mu^C([x_0,y_0])\mathrm{d}\nu_t^{\delta_{[x_0,y_0]}}([x,y]) \left(-\left\|s_t^{\mu^C}([x,y])\right\|^2 + \left\|s_t^{\mu^{Ay_0}}(x)\right\|^2 + \left\|s_t^{\mu^{Bx_0}}(y)\right\|^2\right) \mathrm{d}t =$$

$$\int\limits_0^T \frac{g_t^2}{2} \int \mathrm{d}\mu^C([x_0,y_0])\mathrm{d}\nu_t^{\delta_{[x_0,y_0]}}([x,y]) \left(-\left\|s_t^{\mu^C}([x,y])\right\|^2 + \left\|[s_t^{\mu^{Ay_0}}(x), s_t^{\mu^{Bx_0}}(y)]\right\|^2\right) \mathrm{d}t =$$

$$\int\limits_0^T \frac{g_t^2}{2} \int \mathrm{d}\mu^C([x_0,y_0])\mathrm{d}\nu_t^{\delta_{[x_0,y_0]}}([x,y]) \left(-2\left\|s_t^{\mu^C}([x,y])\right\|^2 + \left\|s_t^{\mu^C}([x,y]) - [s_t^{\mu^{Ay_0}}(x), s_t^{\mu^{Bx_0}}(y)]\right\|^2 +$$

$$2\left\langle s_t^{\mu^C}([x,y]), [s_t^{\mu^{Ay_0}}(x), s_t^{\mu^{Bx_0}}(y)]\right\rangle\right) \mathrm{d}t.$$

Recognizing that the term $\left\|s_t^{\mu^C}([x,y]) - [s_t^{\mu^{Ay_0}}(x), s_t^{\mu^{Bx_0}}(y)]\right\|^2$, averaged over the measures, is just Eq. (21) in disguise, what remain to be assessed is the following:

$$\int\limits_0^T \frac{g_t^2}{2} \int \mathrm{d}\mu^C([x_0,y_0])\mathrm{d}\nu_t^{\delta_{[x_0,y_0]}}([x,y])$$

$$\left(-2\left\|s_t^{\mu^C}([x,y])\right\|^2 + 2\left\langle s_t^{\mu^C}([x,y]), [s_t^{\mu^{Ay_0}}(x), s_t^{\mu^{Bx_0}}(y)]\right\rangle\right) \mathrm{d}t = 0. \quad (24)$$

In particular, we focus on the term:

$$\int\limits_0^T \frac{g_t^2}{2} \int \mathrm{d}\mu^C([x_0,y_0])\mathrm{d}\nu_t^{\delta_{[x_0,y_0]}}([x,y]) \left\langle s_t^{\mu^C}([x,y]), [s_t^{\mu^{Ay_0}}(x), s_t^{\mu^{Bx_0}}(y)]\right\rangle \mathrm{d}t =$$

$$\int\limits_0^T \frac{g_t^2}{2} \int_{x,y} \left\langle s_t^{\mu^C}([x,y]), \right.$$

$$\left[\int_{x_0,y_0} \mathrm{d}\mu^C([x_0,y_0])\mathrm{d}\nu_t^{\delta_{[x_0,y_0]}}([x,y])s_t^{\mu^{Ay_0}}(x), \int_{x_0,y_0} \mathrm{d}\mu^C([x_0,y_0])\mathrm{d}\nu_t^{\delta_{[x_0,y_0]}}([x,y])s_t^{\mu^{Bx_0}}(y)\right]\right\rangle \mathrm{d}t.$$

$$(25)$$

Since $d\nu_t^{\delta_{[x_0,y_0]}}([x,y]) = d\nu_t^{\delta_{x_0}}(x)d\nu_t^{\delta_{y_0}}(y)$ and $d\mu^C([x_0,y_0]) = d\mu^{A_{y_0}}(x_0)d\mu^B(y_0)$, then $\int_{x_0} d\mu^C([x_0,y_0])d\nu_t^{\delta_{[x_0,y_0]}}([x,y]) = d\nu_t^{\mu^{A_{y_0}}}(x)d\nu_t^{\delta_{y_0}}(y)d\mu^B(y_0)$. Consequently:

$$\int_{x_0,y_0} d\mu^C([x_0,y_0])d\nu_t^{\delta_{[x_0,y_0]}}([x,y])s_t^{\mu^{A_{y_0}}}(x) = \int_{y_0} d\nu_t^{\mu^{A_{y_0}}}(x)d\nu_t^{\delta_{y_0}}(y)d\mu^B(y_0)s_t^{\mu^{A_{y_0}}}(x) =$$

$$\int_{y_0} d\nu_t^{\mu^{A_{y_0}}}(x)d\nu_t^{\delta_{y_0}}(y)d\mu^B(y_0)\nabla\log\left(\bar{\nu}_t^{\mu^{A_{y_0}}}(x)\right) = \int_{y_0} d\nu_t^{\mu^{A_{y_0}}}(x)d\nu_t^{\delta_{y_0}}(y)d\mu^B(y_0)\frac{\nabla\bar{\nu}_t^{\mu^{A_{y_0}}}(x)}{\bar{\nu}_t^{\mu^{A_{y_0}}}(x)} =$$

$$dx\int_{y_0} d\nu_t^{\delta_{y_0}}(y)d\mu^B(y_0)\nabla\bar{\nu}_t^{\mu^{A_{y_0}}}(x) = dx\nabla\left(\int_{y_0} d\nu_t^{\delta_{y_0}}(y)d\mu^B(y_0)\bar{\nu}_t^{\mu^{A_{y_0}}}(x)\right) =$$

$$dxd\nu_t^{\mu^B}(y)\nabla\left(\int_{y_0} d\mu^{B\,|\,Y_t=y}(y_0)\bar{\nu}_t^{\mu^{A_{y_0}}}(x)\right) = dxd\nu_t^{\mu^B}(y)\nabla\left(\bar{\nu}_t^{\mu^{A\,|\,Y_t=y}}(x)\right),$$

where in the last line we introduced: $\mu^{B\,|\,Y_t=y}(y_0)$, the measure of the random variable $B$ conditionally on the fact that the diffused variable $B$ after a time $t$ is equal to $y$ and $\nu^{\mu^{A\,|\,Y_t=y}}$, the conditional measure of the diffused variable $A$ at time $t$, conditionally on the diffused variable $B$ after a time $t$ equal to $y$. Finally

$$dxd\nu_t^{\mu^B}(y)\nabla\left(\bar{\nu}_t^{\mu^{A\,|\,Y_t=y}}(x)\right) = \bar{\nu}_t^{\mu^{A\,|\,Y_t=y}}(x)dxd\nu_t^{\mu^B}(y)\frac{\nabla\left(\bar{\nu}_t^{\mu^{A\,|\,Y_t=y}}(x)\right)}{\bar{\nu}_t^{\mu^{A\,|\,Y_t=y}}(x)} = d\nu_t^{\mu^C}([x,y])s_t^{\mu^{A\,|\,Y_t=y}}(x).$$

Along the same lines, we can prove the equality $\int_{x_0,y_0} d\mu^C([x_0,y_0])d\nu_t^{\delta_{[x_0,y_0]}}([x,y])s_t^{\mu^{B_{x_0}}}(y) = d\nu_t^{\mu^C}([x,y])s_t^{\mu^{B\,|\,X_t=x}}(y)$. Then, restarting from Eq. (25) we have:

$$\int_0^T \frac{g_t^2}{2}\int_{x,y}\left\langle s_t^{\mu^C}([x,y]),\right.$$

$$\left.\left[\int_{x_0,y_0} d\mu^C([x_0,y_0])d\nu_t^{\delta_{[x_0,y_0]}}([x,y])s_t^{\mu^{A_{y_0}}}(x), \int_{x_0,y_0} d\mu^C([x_0,y_0])d\nu_t^{\delta_{[x_0,y_0]}}([x,y])s_t^{\mu^{B_{x_0}}}(y)\right]\right\rangle dt =$$

$$\int_0^T \frac{g_t^2}{2}\int_{x,y}\left\langle s_t^{\mu^C}([x,y]),\left[d\nu_t^{\mu^C}([x,y])s_t^{\mu^{A\,|\,Y_t=y}}(x),d\nu_t^{\mu^C}([x,y])s_t^{\mu^{B\,|\,X_t=x}}(y)\right]\right\rangle dt =$$

$$\int_0^T \frac{g_t^2}{2}\int_{x,y} d\nu_t^{\mu^C}([x,y])\left\langle s_t^{\mu^C}([x,y]),[s_t^{\mu^{A\,|\,Y_t=y}}(x),s_t^{\mu^{B\,|\,X_t=x}}(y)]\right\rangle dt = \int_0^T \frac{g_t^2}{2}\int_{x,y} d\nu_t^{\mu^C}([x,y])\left\|s_t^{\mu^C}([x,y])\right\|^2 dt,$$

which finally allows to prove Eq. (24) and claim validity of Eq. (21).

## C IMPLEMENTATION DETAILS

In this Section, we provide additional technical details of MINDE. We discuss the different variants of our method their implementation details, including detailed information about the MI estimators alternatives considered in § 5.

### C.1 MINDE-C

In all experiments, we consider the first variable as the main variable and the second variable as the conditioning signal. A single neural network is used to model the conditional and unconditional score. It accepts as inputs the two variables, the diffusion time $t$, and an additionally binary input $c$ which enable the conditional mode. To enable the conditional mode, we set $c = 1$ and feed the network with both the main variable and the conditioning signal, obtaining $\tilde{s}_t^{\mu^{A_{Y_0}}}$. To obtain the marginal score $\tilde{s}_t^{\mu^A}$, we set $c = 0$ and the conditioning signal is set to zero value.

---

**Algorithm 1:** MINDE–C (Single Training Step)

---

**Data:** $[X_0, Y_0] \sim \mu^C$
**parameter :** $net_\theta()$, with $\theta$ current parameters
$t \sim \mathcal{U}[0, T]$      // Importance sampling can be used to reduce variance
$X_t \leftarrow k_t X_0 + \left(k_t^2 \int_0^t k_s^{-2} g_s^2 \mathrm{d}s\right)^{\frac{1}{2}} \epsilon$, with $\epsilon \sim \gamma_1$      // r.h.s. of Eq. (16), diffuse
  the variable $X$ to timestep $t$
$c \sim \mathrm{Bernoulli}(d)$      // Sample binary variable $c$ with probability $d$
**if** $c = 0$ **then**
    $\dfrac{\hat{\epsilon}}{\left(k_t^2 \int_0^t k_s^{-2} g_s^2 \mathrm{d}s\right)^{\frac{1}{2}}} \leftarrow net_\theta([X_t, 0], t, c = 0)$      // Estimated unconditional score
**else**
    $\dfrac{\hat{\epsilon}}{\left(k_t^2 \int_0^t k_s^{-2} g_s^2 \mathrm{d}s\right)^{\frac{1}{2}}} \leftarrow net_\theta([X_t, Y_0], t, c = 1))$      // Estimated conditional score
$L = \dfrac{g_t^2}{\left(k_t^2 \int_0^t k_s^{-2} g_s^2 \mathrm{d}s\right)} \|\epsilon - \hat{\epsilon}\|^2$      // Compute Montecarlo sample associated to
  Eq. (5)
**return** Update $\theta$ according to gradient of $L$

---

**Algorithm 2:** MINDE–C

---

**Data:** $[X_0, Y_0] \sim \mu^C$
**parameter :** $\sigma, option$
$t \sim \mathcal{U}[0, T]$      // Importance sampling can be used to reduce variance
$X_t \leftarrow k_t X_0 + \left(k_t^2 \int_0^t k_s^{-2} g_s^2 \mathrm{d}s\right)^{\frac{1}{2}} \epsilon$, with $\epsilon \sim \gamma_1$      // r.h.s. of Eq. (16), diffuse
  the variable $X$ to timestep $t$
$\tilde{s}_t^{\mu^A} \leftarrow net_\theta([X_t, 0], t, c = 0)$      // Use the unique score network to compute
$\tilde{s}_t^{\mu^{A_{Y_0}}} \leftarrow net_\theta([X_t, Y_0], t, c = 1))$      // marginal and conditional scores
**if** $option = 1$ **then**
    $\hat{I} \leftarrow T \dfrac{g_t^2}{2} \left\| \tilde{s}_t^{\mu^A} - \tilde{s}_t^{\mu^{A_{Y_0}}} \right\|^2$
**else**
    $\chi_t \leftarrow \left(k_t^2 \sigma^2 + k_t^2 \int_0^t k_s^{-2} g_s^2 \mathrm{d}s\right)$
    $\hat{I} \leftarrow T \dfrac{g_t^2}{2} \left[ \left\| \tilde{s}_t^{\mu^A} + \dfrac{X_t}{\chi_t} \right\|^2 - \left\| \tilde{s}_t^{\mu^{A_{Y_0}}} + \dfrac{X_t}{\chi_t} \right\|^2 \right]$
**return** $\hat{I}$

---

A randomized procedure is used for training. For each training step, with probability $d$, the main variable is diffused and the score network is fed with the diffused variable, the conditioning variable, the diffusion time signal and the conditioning signal is set to $c = 1$. On the contrary, with probability $1 - d$, to enable the network to learn the unconditional score, the network is fed only with the diffused modality, the diffusion time and $c = 0$. In contrast to the first case, the conditioning is not provided to the score network and replaced with a zero value vector. Pseudocode is presented in Algorithm 1.

Actual estimation of the MI is then possible either by leveraging Eq. (18) or Eq. (19), referred to in the main text as difference *outside* or *inside* the score respectively (MINDE-C($\sigma$), MINDE-C). A pseudo-code description is provided in Algorithm 2.

## C.2 MINDE-J

The joint variant of our method, MINDE-J is based on the parametrized joint processes in Eq. (17). Also in this case, instead of training a separate score network for each possible combination of conditional modalities, we use a single architecture that accepts both variables, the diffusion time $t$ and the coefficients $\alpha, \beta$. This approach allows modelling the joint score network $\tilde{s}_t^{\mu^C}$ by setting

$\alpha = \beta = 1$. Similarly, to obtain the conditional scores it is sufficient to set $\alpha = 1, \beta = 0$ or $\alpha = 0, \beta = 1$, corresponding to $\tilde{s}_t^{\mu_{Y_0}^A}$ and $\tilde{s}_t^{\mu_{X_0}^A}$ respectively.

Training is carried out again through a randomized procedure. At each training step, with probability $d$, both variables are diffused. In this case, the score network is fed with diffusion time $t$, along with $X_t, Y_t$ and the two parameters $\alpha = \beta = 1$. With probability $1 - d$, instead, we randomly select one variable to be diffused, while we keeping constant the other. For instance, if $A$ is the one which is diffused, we set $\alpha = 1$ and $\beta = 0$. Further details are presented in Algorithm 3.

Once the score network is trained, MI estimation can be obtained following the procedure explained in Algorithm 4. Two options are possible, either by computing the difference between the parametric scores outside the same norm (Eq. (20) MINDE-J($\sigma$)) or inside (Eq. (21) MINDE-J). Similarly to the conditional case, an *option* parameter can be used to switch among the two.

---

**Algorithm 3:** MINDE–J (Single Training Step)

---

**Data:** $[X_0, Y_0] \sim \mu^C$
**parameter :** $net_\theta()$, with $\theta$ current parameters
$t \sim \mathcal{U}[0, T]$       `// Importance sampling can be used to reduce variance`

$[X_t, Y_t] \leftarrow k_t[X_0, Y_0] + \left( k_t^2 \int_0^t k_s^{-2} g_s^2 \mathrm{d}s \right)^{\frac{1}{2}} [\epsilon_1, \epsilon_2]$, with $\epsilon_{1,2} \sim \gamma_1$ `// l.h.s.` `Eq. (16),`
`diffuse modalities to timestep` $t$
$c \sim \text{Bernoulli}(d)$       `// Sample binary variable c with probability d`
**if** $c = 0$ **then**

> $\dfrac{[\hat{\epsilon}_1, \hat{\epsilon}_2]}{\left( k_t^2 \int_0^t k_s^{-2} g_s^2 \mathrm{d}s \right)^{\frac{1}{2}}} \leftarrow net_\theta([X_t, Y_t], t, [1, 1])$    `// Estimated unconditional score`
>
> $L = \dfrac{g_t^2}{\left( k_t^2 \int_0^t k_s^{-2} g_s^2 \mathrm{d}s \right)} \| [\epsilon_1, \epsilon_2] - [\hat{\epsilon}_1, \hat{\epsilon}_2] \|^2$       `// Compute Montecarlo sample`
> `associated to` `Eq. (5)`

**else**

> **if** *Bernoulli(0.5)* **then**
>
>> $\dfrac{\hat{\epsilon}_1}{\left( k_t^2 \int_0^t k_s^{-2} g_s^2 \mathrm{d}s \right)^{\frac{1}{2}}} \leftarrow net_\theta([X_t, Y_0], t, [1, 0])$    `// Estimated Conditional score`
>
> $L = \dfrac{g_t^2}{\left( k_t^2 \int_0^t k_s^{-2} g_s^2 \mathrm{d}s \right)} \| \epsilon_1 - \hat{\epsilon}_1 \|^2$
>
> **else**
>
>> $\dfrac{\hat{\epsilon}_2}{\left( k_t^2 \int_0^t k_s^{-2} g_s^2 \mathrm{d}s \right)^{\frac{1}{2}}} \leftarrow net_\theta([X_0, Y_t], t, [0, 1])$    `// Estimated Conditional score`
>
> $L = \dfrac{g_t^2}{\left( k_t^2 \int_0^t k_s^{-2} g_s^2 \mathrm{d}s \right)} \| \epsilon_2 - \hat{\epsilon}_2 \|^2$

**return** Update $\theta$ according to gradient of $L$

---

### C.3    Technical settings for MINDE-C and MINDE-J

We follow the implementation of Bounoua et al. [2023] which uses stacked multi-layer perception (MLP) with skip connections. We adopt a simplified version of the same score network architecture: this involves three Residual MLP blocks. We use the *Adam optimizer* [Kingma & Ba, 2015] for training and Exponential moving average (EMA) with a momentum parameter $m = 0.999$. We use importance sampling at train and test-time. We returned the mean estimate on the test data set over 10 runs.

The hyper-parameters are presented in Table 2 and Table 3 for MINDE-J and MINDE-C respectively. Concerning the consistency tests (§ 5.2), we independently train an autoencoder for each version of the MNIST dataset with $r$ rows available.

---

**Algorithm 4:** MINDE–J

---

**Data:** $[X_0, Y_0] \sim \mu^C$
**parameter :** $\sigma$, *option*
$t \sim \mathcal{U}[0, T]$        `// Importance sampling can be used to reduce variance`
$[X_t, Y_t] \leftarrow k_t[X_0, Y_0] + \left(k_t^2 \int_0^t k_s^{-2} g_s^2 \mathrm{d}s\right)^{\frac{1}{2}} [\epsilon_1, \epsilon_2], \text{ with } \epsilon_{1,2} \sim \gamma_1$ `// l.h.s.` `Eq. (16)`, `diffuse modalities to timestep t`
$\tilde{s}_t^{\mu^C} \leftarrow net_\theta([X_t, Y_t], t, [1, 1])$       `// Use the unique score network to compute joint`
$\tilde{s}_t^{\mu_{Y_0}^A} \leftarrow net_\theta([X_t, Y_0], t, [1, 0])$             `// and conditional scores`
$\tilde{s}_t^{\mu_{X_0}^A} \leftarrow net_\theta([X_0, Y_t], t, [0, 1])$
**if** *option* = 1 **then**

$\quad \hat{I} \leftarrow T\frac{g_t^2}{2} \left\| \tilde{s}_t^{\mu^C} - [\tilde{s}_t^{\mu^{A_{Y_0}}}, \tilde{s}_t^{\mu^{B_{X_0}}}] \right\|^2$

**else**

$\quad \chi_t \leftarrow \left(k_t^2 \sigma^2 + k_t^2 \int_0^t k_s^{-2} g_s^2 \mathrm{d}s\right)$

$\quad \hat{I} \leftarrow T\frac{g_t^2}{2} \left[ \left\| \tilde{s}_t^{\mu^C} + \frac{[X_t, Y_t]}{\chi_t} \right\|^2 - \left\| \tilde{s}_t^{\mu^{A_{Y_0}}} + \frac{X_t}{\chi_t} \right\|^2 - \left\| \tilde{s}_t^{\mu^{B_{X_0}}} + \frac{Y_t}{\chi_t} \right\|^2 \right]$

**return** $\hat{I}$

---

**Table 2:** MINDE-J score network training hyper-parameters. $Dim$ of the task correspond the sum of the two variables dimensions, whereas $d$ corresponds to the randomization probability.

| | $d$ | Width | Time embed | Batch size | Lr | Iterations | Number of params |
|---|---|---|---|---|---|---|---|
| Benchmark ($Dim \leq 10$) | 0.5 | 64 | 64 | 128 | 1e-3 | 234k | 55490 |
| Benchmark ($Dim = 50$) | 0.5 | 128 | 128 | 256 | 2e-3 | 195k | 222100 |
| Benchmark ($Dim = 100$) | 0.5 | 256 | 256 | 256 | 2e-3 | 195k | 911204 |
| Consistency tests | 0.5 | 256 | 256 | 64 | 1e-3 | 390k | 1602080 |

### C.4 NEURAL ESTIMATORS IMPLEMENTATION

We use the package *benchmark-mi*[1] implementation to study the neural estimators. We use MLP architecture with 3 layers of the same width as in MINDE. We use the same training procedure as in Czyż et al. [2023], including early stopping strategy. We return the highest estimate on the test data.

## D ABLATIONS STUDY

### D.1 $\sigma$ ABLATION STUDY

We hereafter report in Table 4 the results of all the variants of MINDE, including different values of $\sigma$ parameter. For completeness in our experimental campaign, we report also the results of non neural competitors, similarly to the work in Czyż et al. [2023]. In summary, the MINDE-C/J versions ("*difference inside*") of our estimator prove to be more robust than the MINDE-C/J($\sigma$) ("*difference outside*") counterpart, especially for the joint variants. Nevertheless, it is interesting to notice that the "*difference outside*" variants are stable and competitive against a very wide range of values of $\sigma$ (ranging from 0.5 to 10), with their best value typically achieved for $\sigma = 1.0$.

---

[1] https://github.com/cbg-ethz/bmi

**Table 3:** MINDE-C score network training hyper-parameters. $Dim$ of the task correspond the sum of the two variables dimensions, and $d$ corresponds to the randomization probability.

| | $d$ | Width | Time embed | Batch size | Lr | Iterations | Number of params |
|---|---|---|---|---|---|---|---|
| Benchmark ($Dim \leq 10$) | 0.5 | 64 | 64 | 128 | 1e-3 | 390k | 55425 |
| Benchmark ($Dim = 50$) | 0.5 | 128 | 128 | 256 | 2e-3 | 290k | 220810 |
| Benchmark ($Dim = 100$) | 0.5 | 256 | 256 | 256 | 2e-3 | 290k | 898354 |
| Consistency tests | 0.5 | 256 | 256 | 64 | 1e-3 | 390k | 1597968 |

| GT | Asinh @ St 1×1 (dof=1) | Asinh @ St 2×2 (dof=1) | Asinh @ St 3×3 (dof=2) | Asinh @ St 5×5 (dof=2) | Bimodal 1×1 | Bivariate Nm 1×1 | Hc @ Bivariate Nm 1×1 | Hc @ Mn 25×25 (2-pair) | Hc @ Mn 3×3 (2-pair) | Hc @ Mn 5×5 (2-pair) | Mn 2×2 (2-pair) | Mn 2×2 (dense) | Mn 25×25 (2-pair) | Mn 25×25 (dense) | Mn 3×3 (2-pair) | Mn 3×3 (dense) | Mn 5×5 (2-pair) | Mn 5×5 (dense) | Mn 50×50 (dense) | Nm CDF @ Bivariate Nm 1×1 | Nm CDF @ Mn 25×25 (2-pair) | Nm CDF @ Mn 3×3 (2-pair) | Nm CDF @ Mn 5×5 (2-pair) | Sp @ Mn 25×25 (2-pair) | Sp @ Mn 3×3 (2-pair) | Sp @ Mn 5×5 (2-pair) | Sp @ Nm CDF @ Mn 25×25 (2-pair) | Sp @ Nm CDF @ Mn 3×3 (2-pair) | Sp @ Nm CDF @ Mn 5×5 (2-pair) | St 1×1 (dof=1) | St 2×2 (dof=1) | St 2×2 (dof=2) | St 3×3 (dof=2) | St 3×3 (dof=3) | St 5×5 (dof=2) | St 5×5 (dof=3) | Swiss roll 2×1 | Uniform 1×1 (additive noise=0.1) | Uniform 1×1 (additive noise=0.75) | Wiggly @ Bivariate Nm 1×1 |
|---|---|---|---|---|---|---|---|---|---|---|---|---|---|---|---|---|---|---|---|---|---|---|---|---|---|---|---|---|---|---|---|---|---|---|---|---|---|---|---|---|
| **MINDE-j** | 0.2 | 0.4 | 0.3 | 0.4 | 0.4 | 0.4 | 0.4 | 1.2 | 1.0 | 1.0 | 0.3 | 1.0 | 1.3 | 1.0 | 1.0 | 0.4 | 1.0 | 0.6 | 1.7 | 0.4 | 1.1 | 1.0 | 1.0 | 0.9 | 1.0 | 1.0 | 0.9 | 1.0 | 1.0 | 0.1 | 0.3 | 0.2 | 0.3 | 0.2 | 0.5 | 0.3 | 0.5 | 1.7 | 0.3 | 0.4 |
| **MINDE-c** | 0.2 | 0.4 | 0.3 | 0.4 | 0.4 | 0.4 | 0.4 | 1.0 | 1.0 | 1.0 | 0.3 | 1.0 | 1.3 | 1.0 | 1.0 | 0.4 | 1.0 | 0.6 | 1.6 | 0.4 | 1.0 | 1.0 | 1.0 | 0.9 | 1.0 | 1.0 | 0.9 | 1.0 | 1.0 | 0.1 | 0.3 | 0.2 | 0.3 | 0.2 | 0.4 | 0.3 | 0.4 | 1.7 | 0.3 | 0.4 |
| **MINDE-j ($\sigma = 0.5$)** | 0.2 | 0.3 | 0.3 | 0.3 | 0.4 | 0.4 | 0.4 | 1.3 | 1.0 | 1.1 | 0.3 | 1.0 | 1.1 | 1.0 | 1.0 | 0.4 | 1.0 | 0.6 | 1.7 | 0.4 | 0.9 | 1.0 | 0.9 | 0.8 | 0.9 | 0.9 | 0.9 | 1.0 | 1.0 | 0.2 | 0.3 | 0.2 | 0.3 | 0.2 | 0.5 | 0.3 | 0.5 | 1.7 | 0.3 | 0.4 |
| **MINDE-j ($\sigma = 1$)** | 0.2 | 0.3 | 0.3 | 0.3 | 0.4 | 0.4 | 0.4 | 1.1 | 1.0 | 1.0 | 0.3 | 1.0 | 1.2 | 1.0 | 1.0 | 0.4 | 1.0 | 0.6 | 1.7 | 0.4 | 0.9 | 1.0 | 0.9 | 0.9 | 1.0 | 1.0 | 0.9 | 1.0 | 1.0 | 0.2 | 0.4 | 0.2 | 0.3 | 0.2 | 0.4 | 0.3 | 0.5 | 1.6 | 0.3 | 0.4 |
| **MINDE-j ($\sigma = 1.5$)** | 0.2 | 0.3 | 0.3 | 0.3 | 0.4 | 0.4 | 0.4 | 1.0 | 1.0 | 1.0 | 0.3 | 1.0 | 1.2 | 1.0 | 1.0 | 0.4 | 1.0 | 0.6 | 1.7 | 0.4 | 0.9 | 1.0 | 0.9 | 0.9 | 1.0 | 1.0 | 0.9 | 1.0 | 1.0 | 0.2 | 0.4 | 0.2 | 0.2 | 0.2 | 0.5 | 0.3 | 0.5 | 1.6 | 0.3 | 0.4 |
| **MINDE-j ($\sigma = 2$)** | 0.2 | 0.3 | 0.3 | 0.4 | 0.4 | 0.4 | 0.4 | 1.0 | 0.9 | 1.0 | 0.3 | 1.0 | 1.3 | 1.0 | 1.0 | 0.4 | 0.9 | 0.6 | 1.8 | 0.4 | 0.9 | 1.0 | 0.8 | 0.9 | 1.0 | 1.0 | 0.9 | 1.0 | 1.0 | 0.2 | 0.4 | 0.2 | 0.2 | 0.1 | 0.5 | 0.3 | 0.5 | 1.7 | 0.3 | 0.4 |
| **MINDE-j ($\sigma = 3$)** | 0.2 | 0.2 | 0.2 | 0.4 | 0.4 | 0.4 | 0.4 | 0.9 | 0.9 | 1.0 | 0.3 | 1.0 | 1.3 | 1.0 | 1.0 | 0.3 | 0.9 | 0.6 | 1.8 | 0.4 | 0.9 | 1.0 | 0.8 | 0.8 | 0.9 | 0.9 | 0.8 | 0.9 | 0.9 | 0.2 | 0.4 | 0.2 | 0.2 | 0.1 | 0.5 | 0.2 | 0.5 | 1.6 | 0.3 | 0.4 |
| **MINDE-j ($\sigma = 5$)** | 0.2 | 0.4 | 0.2 | 0.4 | 0.4 | 0.4 | 0.4 | 0.9 | 0.9 | 1.0 | 0.3 | 1.0 | 1.4 | 1.0 | 1.0 | 0.4 | 0.9 | 0.6 | 1.8 | 0.4 | 0.8 | 0.8 | 0.9 | 0.9 | 1.0 | 1.0 | 0.8 | 0.9 | 0.9 | 0.2 | 0.4 | 0.2 | 0.2 | 0.1 | 0.5 | 0.2 | 0.5 | 1.6 | 0.3 | 0.4 |
| **MINDE-j ($\sigma = 10$)** | 0.2 | 0.2 | 0.2 | 0.4 | 0.3 | 0.4 | 0.4 | 0.8 | 0.9 | 1.0 | 0.3 | 1.0 | 1.4 | 1.0 | 1.0 | 0.4 | 0.9 | 0.6 | 1.9 | 0.4 | 0.8 | 0.9 | 0.8 | 0.8 | 0.9 | 0.9 | 0.8 | 0.9 | 0.9 | 0.2 | 0.4 | 0.2 | 0.2 | 0.1 | 0.5 | 0.2 | 0.5 | 1.6 | 0.3 | 0.4 |
| **MINDE-c ($\sigma = 0.5$)** | 0.2 | 0.3 | 0.3 | 0.3 | 0.4 | 0.4 | 0.4 | 0.9 | 1.0 | 1.0 | 0.3 | 1.0 | 1.3 | 1.0 | 1.0 | 0.4 | 1.0 | 0.6 | 1.7 | 0.4 | 0.9 | 1.0 | 0.9 | 0.8 | 0.9 | 0.9 | 0.9 | 1.0 | 1.0 | 0.1 | 0.3 | 0.2 | 0.3 | 0.2 | 0.4 | 0.3 | 0.4 | 1.6 | 0.3 | 0.4 |
| **MINDE-c ($\sigma = 1$)** | 0.2 | 0.3 | 0.3 | 0.3 | 0.4 | 0.4 | 0.4 | 1.0 | 1.0 | 1.0 | 0.3 | 1.0 | 1.3 | 1.0 | 1.0 | 0.4 | 1.0 | 0.6 | 1.6 | 0.4 | 0.9 | 1.0 | 0.9 | 0.9 | 1.0 | 1.0 | 0.9 | 1.0 | 1.0 | 0.1 | 0.3 | 0.2 | 0.3 | 0.2 | 0.4 | 0.3 | 0.4 | 1.6 | 0.3 | 0.4 |
| **MINDE-c ($\sigma = 1.5$)** | 0.2 | 0.3 | 0.3 | 0.3 | 0.4 | 0.4 | 0.4 | 1.0 | 1.0 | 1.0 | 0.3 | 1.0 | 1.3 | 1.0 | 1.0 | 0.4 | 1.0 | 0.6 | 1.6 | 0.4 | 0.9 | 1.0 | 0.9 | 0.9 | 1.0 | 1.0 | 0.9 | 1.0 | 1.0 | 0.1 | 0.3 | 0.2 | 0.3 | 0.2 | 0.4 | 0.3 | 0.4 | 1.6 | 0.3 | 0.4 |
| **MINDE-c ($\sigma = 2$)** | 0.2 | 0.3 | 0.3 | 0.4 | 0.4 | 0.4 | 0.4 | 1.0 | 0.9 | 1.0 | 0.3 | 1.0 | 1.2 | 1.0 | 1.0 | 0.4 | 0.9 | 0.6 | 1.6 | 0.4 | 0.9 | 1.0 | 0.9 | 0.9 | 1.0 | 1.0 | 0.9 | 1.0 | 1.0 | 0.1 | 0.3 | 0.2 | 0.3 | 0.1 | 0.4 | 0.3 | 0.4 | 1.6 | 0.3 | 0.4 |
| **MINDE-c ($\sigma = 3$)** | 0.2 | 0.3 | 0.3 | 0.4 | 0.4 | 0.4 | 0.4 | 1.0 | 0.9 | 1.0 | 0.3 | 1.0 | 1.3 | 1.0 | 1.0 | 0.4 | 0.9 | 0.6 | 1.6 | 0.4 | 0.9 | 1.0 | 0.9 | 0.9 | 1.0 | 1.0 | 0.9 | 1.0 | 1.0 | 0.1 | 0.3 | 0.2 | 0.3 | 0.1 | 0.4 | 0.3 | 0.4 | 1.7 | 0.3 | 0.4 |
| **MINDE-c ($\sigma = 5$)** | 0.2 | 0.4 | 0.3 | 0.4 | 0.4 | 0.4 | 0.4 | 1.0 | 0.9 | 1.0 | 0.3 | 1.0 | 1.2 | 1.0 | 1.0 | 0.4 | 0.9 | 0.6 | 1.8 | 0.4 | 0.8 | 0.8 | 0.9 | 0.9 | 1.0 | 1.0 | 0.9 | 1.0 | 1.0 | 0.1 | 0.3 | 0.2 | 0.3 | 0.1 | 0.4 | 0.2 | 0.4 | 1.7 | 0.3 | 0.4 |
| **MINDE-c ($\sigma = 10$)** | 0.2 | 0.2 | 0.2 | 0.4 | 0.3 | 0.4 | 0.4 | 0.8 | 0.9 | 1.0 | 0.3 | 1.0 | 1.2 | 1.0 | 1.0 | 0.4 | 0.8 | 0.6 | 1.6 | 0.4 | 0.8 | 0.8 | 0.8 | 0.8 | 1.0 | 1.0 | 0.9 | 1.0 | 1.0 | 0.1 | 0.3 | 0.2 | 0.3 | 0.2 | 0.4 | 0.3 | 0.4 | 1.7 | 0.3 | 0.4 |
| **MINE** | 0.2 | 0.4 | 0.3 | 0.4 | 0.4 | 0.4 | 0.4 | 1.0 | 1.0 | 1.0 | 0.3 | 1.0 | 1.3 | 1.0 | 1.0 | 0.4 | 1.0 | 0.6 | 1.6 | 0.4 | 0.9 | 0.9 | 0.8 | 0.8 | 0.9 | 0.8 | 0.9 | 0.9 | 0.9 | 0.0 | 0.1 | 0.1 | 0.1 | 0.1 | 0.2 | 0.2 | 0.4 | 1.7 | 0.3 | 0.4 |
| **InfoNCE** | 0.2 | 0.4 | 0.3 | 0.4 | 0.4 | 0.4 | 0.4 | 1.0 | 1.0 | 1.0 | 0.3 | 1.0 | 1.3 | 1.0 | 1.0 | 0.4 | 1.0 | 0.6 | 1.6 | 0.4 | 1.0 | 1.0 | 0.8 | 0.8 | 0.8 | 0.8 | 1.0 | 1.0 | 0.7 | 0.2 | 0.2 | 0.2 | 0.3 | 0.2 | 0.4 | 0.3 | 0.4 | 1.7 | 0.3 | 0.4 |
| **D-V** | 0.2 | 0.4 | 0.3 | 0.4 | 0.4 | 0.4 | 0.4 | 1.0 | 1.0 | 1.0 | 0.3 | 1.0 | 1.3 | 1.0 | 1.0 | 0.4 | 1.0 | 0.6 | 1.6 | 0.4 | 0.9 | 1.0 | 0.8 | 0.8 | 0.8 | 0.8 | 0.9 | 1.0 | 1.0 | 0.0 | 0.0 | 0.0 | 0.1 | 0.0 | 0.1 | 0.2 | 0.4 | 1.7 | 0.3 | 0.4 |
| **NWJ** | 0.2 | 0.4 | 0.3 | 0.4 | 0.4 | 0.4 | 0.4 | 1.0 | 1.0 | 1.0 | 0.3 | 1.0 | 1.3 | 1.0 | 1.0 | 0.4 | 1.0 | 0.6 | 1.6 | 0.4 | 0.9 | 1.0 | 0.8 | 0.8 | 0.8 | 0.8 | 0.9 | 1.0 | 1.0 | 0.0 | 0.0 | 0.0 | -0.6 | 0.1 | 0.2 | 0.1 | 0.4 | 1.7 | 0.3 | 0.4 |
| **KSG** | 0.2 | 0.9 | 0.2 | 0.2 | 0.4 | 0.4 | 0.2 | 0.2 | 0.9 | 0.9 | 0.2 | 0.9 | 0.2 | 1.1 | 1.0 | 0.4 | 0.7 | 0.6 | 1.3 | 0.2 | 0.2 | 0.9 | 0.7 | 0.2 | 0.8 | 0.6 | 0.2 | 0.9 | 0.7 | 0.2 | 0.2 | 0.1 | 0.1 | 0.1 | 0.2 | 0.2 | 0.3 | 1.7 | 0.3 | 0.4 |
| **LNN** | 2.7 | 6.6 | 2.7 | 5.5 | 0.3 | 0.4 | 17.3 | 17.3 | 2.5 | 7.3 | 0.6 | 17.3 | 17.3 | 17.3 | 3.1 | 2.5 | 7.3 | 6.8 | 33.5 | 17.3 | 17.3 | 2.4 | 7.2 | 0.9 | 3.1 | 7.3 | 17.3 | 2.5 | 7.3 | -0.2 | 0.5 | 0.2 | 1.2 | 2.1 | 2.7 | 5.5 | 0.0 | 1.5 | 0.3 | 0.4 |
| **CCA** | 0.0 | 0.0 | 0.0 | 0.0 | 0.3 | 0.4 | 1.0 | 17.3 | 1.0 | 1.0 | 0.3 | 1.1 | 17.3 | 1.3 | 1.0 | 0.4 | 1.0 | 0.6 | 1.8 | 1.0 | 17.3 | 0.8 | 0.4 | 0.9 | 0.2 | 0.4 | 0.9 | 0.8 | 0.4 | 0.3 | 0.1 | 0.1 | 0.1 | 0.0 | 0.3 | 0.0 | 0.0 | 1.6 | 0.2 | 0.4 |

**Table 4:** MINDE-j and MINDE-c $\sigma$ ablations study. Mean MI estimates over 10 seeds using N = 10000 samples compared each against the ground-truth. Color indicates relative negative bias (red) and positive bias (blue). Our method and all neural estimators were trained with 100k training samples. List of abbreviations ( *Mn*: Multinormal, *St*: Student-t, *Nm*: Normal, *Hc*: Half-cube, *Sp*: Spiral)

## D.2    FULL RESULTS WITH STANDARD DEVIATION

We report in Table 5 mean results without quantization for the different methods. Figures 3 and 4 contains box-plots for all the competitors and all the tasks.

| | Asinh @ St 1×1 (dof=1) | Asinh @ St 2×2 (dof=1) | Asinh @ St 3×3 (dof=2) | Asinh @ St 5×5 (dof=2) | Bimodal 1×1 | Bivariate Nm 1×1 | Hc @ Bivariate Nm 1×1 | Hc @ Mn 25×25 (2-pair) | Hc @ Mn 3×3 (2-pair) | Hc @ Mn 5×5 (2-pair) | Mn 2×2 (2-pair) | Mn 2×2 (dense) | Mn 25×25 (2-pair) | Mn 25×25 (dense) | Mn 3×3 (2-pair) | Mn 3×3 (dense) | Mn 5×5 (2-pair) | Mn 5×5 (dense) | Mn 50×50 (dense) | Nm CDF @ Bivariate Nm 1×1 | Nm CDF @ Mn 25×25 (2-pair) | Nm CDF @ Mn 3×3 (2-pair) | Nm CDF @ Mn 5×5 (2-pair) | Sp @ Mn 25×25 (2-pair) | Sp @ Mn 3×3 (2-pair) | Sp @ Mn 5×5 (2-pair) | Sp @ Nm CDF @ Mn 25×25 (2-pair) | Sp @ Nm CDF @ Mn 3×3 (2-pair) | Sp @ Nm CDF @ Mn 5×5 (2-pair) | St 1×1 (dof=1) | St 2×2 (dof=1) | St 2×2 (dof=2) | St 3×3 (dof=2) | St 3×3 (dof=3) | St 5×5 (dof=2) | St 5×5 (dof=3) | Swiss roll 2×1 | Uniform 1×1 (additive noise=.1) | Uniform 1×1 (additive noise=.75) | Wiggly @ Bivariate Nm 1×1 |
|---|---|---|---|---|---|---|---|---|---|---|---|---|---|---|---|---|---|---|---|---|---|---|---|---|---|---|---|---|---|---|---|---|---|---|---|---|---|---|---|---|
| GT | 0.22 | 0.43 | 0.29 | 0.45 | 0.41 | 0.41 | 1.02 | 1.02 | 0.29 | 1.02 | 1.02 | 0.29 | 1.02 | 1.02 | 0.41 | 1.02 | 1.02 | 1.02 | 1.02 | 0.22 | 0.43 | 0.19 | 0.29 | 0.18 | 0.45 | 0.30 | 0.41 | 1.71 | 0.33 | 0.41 | | | | | | | | | | |
| MINDE-j ($\sigma = 1$) | 0.21 | 0.40 | 0.26 | 0.40 | 0.41 | 0.41 | 1.10 | 1.01 | 0.29 | 0.91 | 1.18 | 1.00 | 0.42 | 1.00 | 0.59 | 1.72 | 0.40 | 0.96 | 0.96 | 0.99 | 0.87 | 0.90 | 0.90 | 0.95 | 0.95 | 0.98 | 0.17 | 0.35 | 0.18 | 0.25 | 0.17 | 0.47 | 0.29 | 0.49 | 1.65 | 0.31 | 0.41 | | | |
| MINDE-j | 0.22 | 0.42 | 0.28 | 0.42 | 0.41 | 0.42 | 1.19 | 1.02 | 0.29 | 0.99 | 1.31 | 1.02 | 0.29 | 1.01 | 0.59 | 1.73 | 0.41 | 1.07 | 0.99 | 0.99 | 0.95 | 0.92 | 0.93 | 1.07 | 0.99 | 0.98 | 0.13 | 0.24 | 0.20 | 0.30 | 0.18 | 0.48 | 0.31 | 0.40 | 1.67 | 0.31 | 0.42 | | | |
| MINDE-c ($\sigma = 1$) | 0.21 | 0.42 | 0.27 | 0.42 | 0.41 | 0.41 | 0.96 | 1.00 | 0.29 | 0.99 | 1.26 | 1.01 | 0.41 | 1.01 | 0.59 | 1.62 | 0.40 | 0.94 | 0.97 | 0.95 | 0.92 | 0.94 | 0.93 | 0.96 | 0.94 | 0.94 | 0.13 | 0.28 | 0.18 | 0.29 | 0.18 | 0.42 | 0.30 | 0.32 | 1.67 | 0.31 | 0.41 | | | |
| MINDE-c | 0.21 | 0.42 | 0.28 | 0.42 | 0.41 | 0.41 | 1.00 | 1.01 | 0.29 | 1.01 | 1.27 | 1.01 | 0.41 | 1.01 | 0.59 | 1.60 | 0.40 | 0.98 | 0.99 | 0.98 | 0.92 | 0.94 | 0.94 | 0.98 | 0.98 | 0.98 | 0.14 | 0.26 | 0.19 | 0.28 | 0.17 | 0.44 | 0.29 | 0.40 | 1.66 | 0.31 | 0.41 | | | |
| MINE | 0.23 | 0.38 | 0.24 | 0.36 | 0.40 | 0.41 | 0.96 | 0.99 | 0.30 | 0.99 | 1.28 | 1.01 | 0.41 | 1.00 | 0.59 | 1.60 | 0.39 | 0.88 | 0.90 | 0.90 | 0.81 | 0.70 | 0.65 | 0.88 | 0.89 | 0.87 | 0.02 | 0.01 | 0.12 | 0.13 | 0.16 | 0.22 | 0.39 | 1.66 | 0.32 | 0.41 | | | | |
| InfoNCE | 0.22 | 0.41 | 0.27 | 0.40 | 0.41 | 0.41 | 0.98 | 1.01 | 0.29 | 0.99 | 1.28 | 1.01 | 0.41 | 1.02 | 0.59 | 1.61 | 0.40 | 0.92 | 0.98 | 0.99 | 0.83 | 0.84 | 0.82 | 0.92 | 0.96 | 0.96 | 0.15 | 0.30 | 0.18 | 0.27 | 0.17 | 0.41 | 0.28 | 0.40 | 1.69 | 0.32 | 0.41 | | | |
| D-V | 0.22 | 0.41 | 0.27 | 0.40 | 0.41 | 0.41 | 0.98 | 1.01 | 0.29 | 0.99 | 1.28 | 1.01 | 0.41 | 1.02 | 0.59 | 1.60 | 0.40 | 0.93 | 0.98 | 0.99 | 0.82 | 0.82 | 0.81 | 0.92 | 0.96 | 0.96 | 0.01 | 0.05 | 0.11 | 0.13 | 0.15 | 0.22 | 0.40 | 1.69 | 0.32 | 0.41 | | | | |
| NWJ | 0.22 | 0.41 | 0.27 | 0.40 | 0.41 | 0.41 | 0.98 | 1.01 | 0.29 | 0.99 | 1.28 | 1.01 | 0.41 | 1.02 | 0.59 | 1.60 | 0.40 | 0.93 | 0.98 | 0.82 | 0.82 | 0.80 | 0.92 | 0.95 | 0.96 | 0.03 | 0.02 | 0.04 | -0.65 | 0.12 | 0.12 | 0.21 | 0.40 | 1.69 | 0.32 | 0.41 | | | | |
| KSG | 0.22 | 0.38 | 0.19 | 0.24 | 0.42 | 0.42 | 0.17 | 0.87 | 0.66 | 1.03 | 0.29 | 0.20 | 1.07 | 0.95 | 0.41 | 0.74 | 0.57 | 1.28 | 0.42 | 0.20 | 0.92 | 0.72 | 0.18 | 0.71 | 0.55 | 0.20 | 0.90 | 0.69 | 0.16 | 0.22 | 0.09 | 0.12 | 0.07 | 0.20 | 0.15 | 0.42 | 1.68 | 0.32 | 0.42 | |
| LNN | 0.25 | 0.89 | 2.71 | 6.65 | 0.41 | 0.42 | 0.42 | 2.68 | 6.45 | 3.10 | 2.49 | 7.31 | 6.76 | 0.39 | 2.49 | 7.27 | 3.10 | 7.31 | 0.39 | 1.20 | 2.12 | 2.74 | 5.48 | 0.34 | 1.48 | 0.29 | 0.42 | | | | | | | | | 0.53 | | | | |
| CCA | 0.00 | 0.00 | 0.00 | 0.00 | 0.34 | 0.41 | 0.38 | 0.99 | 0.95 | 0.96 | 1.02 | 0.29 | 1.06 | 1.33 | 1.02 | 0.41 | 1.02 | 0.60 | 1.75 | 1.00 | 0.97 | 0.96 | 0.86 | 0.23 | 0.39 | 0.33 | 0.95 | 0.11 | 0.13 | 0.01 | 0.31 | 0.02 | 1.63 | 0.19 | 0.38 | | | | |
| DoE(Gaussian) | 0.16 | 0.48 | 0.27 | 0.57 | 0.37 | 0.39 | 0.67 | 0.97 | 0.96 | 0.95 | 0.35 | 0.67 | 7.83 | 0.95 | 0.64 | 0.94 | 1.27 | 16.11 | 0.39 | 1.00 | 0.97 | 0.99 | 0.95 | 0.48 | 0.58 | 0.56 | 0.57 | 0.74 | 0.77 | 6.74 | 7.93 | 1.78 | 2.54 | 0.61 | 4.24 | 1.17 | 1.57 | 0.11 | 0.38 | |
| DoE(Logistic) | 0.13 | 0.37 | 0.21 | 0.43 | 0.35 | 0.41 | 0.62 | 0.92 | 0.92 | 0.95 | 0.34 | 0.69 | 7.83 | 0.95 | 0.63 | 0.93 | 1.27 | 16.15 | 0.41 | 0.78 | 1.08 | 1.05 | 0.47 | 0.60 | 0.55 | 0.67 | 0.79 | 0.81 | -0.32 | 2.00 | 0.46 | 0.82 | 0.29 | 1.48 | 0.59 | 1.58 | 0.08 | 0.35 | | |

**Table 5:** Mean estimate over 10 seeds using N = 10000 samples compared each against the ground-truth. Our method and all neural estimators were trained with 100k training samples. List of abbreviations (*Mn*: Multinomial, *St*: Student-t, *Nm*: Normal, *Hc*: Half-cube, *Sp*: Spiral)

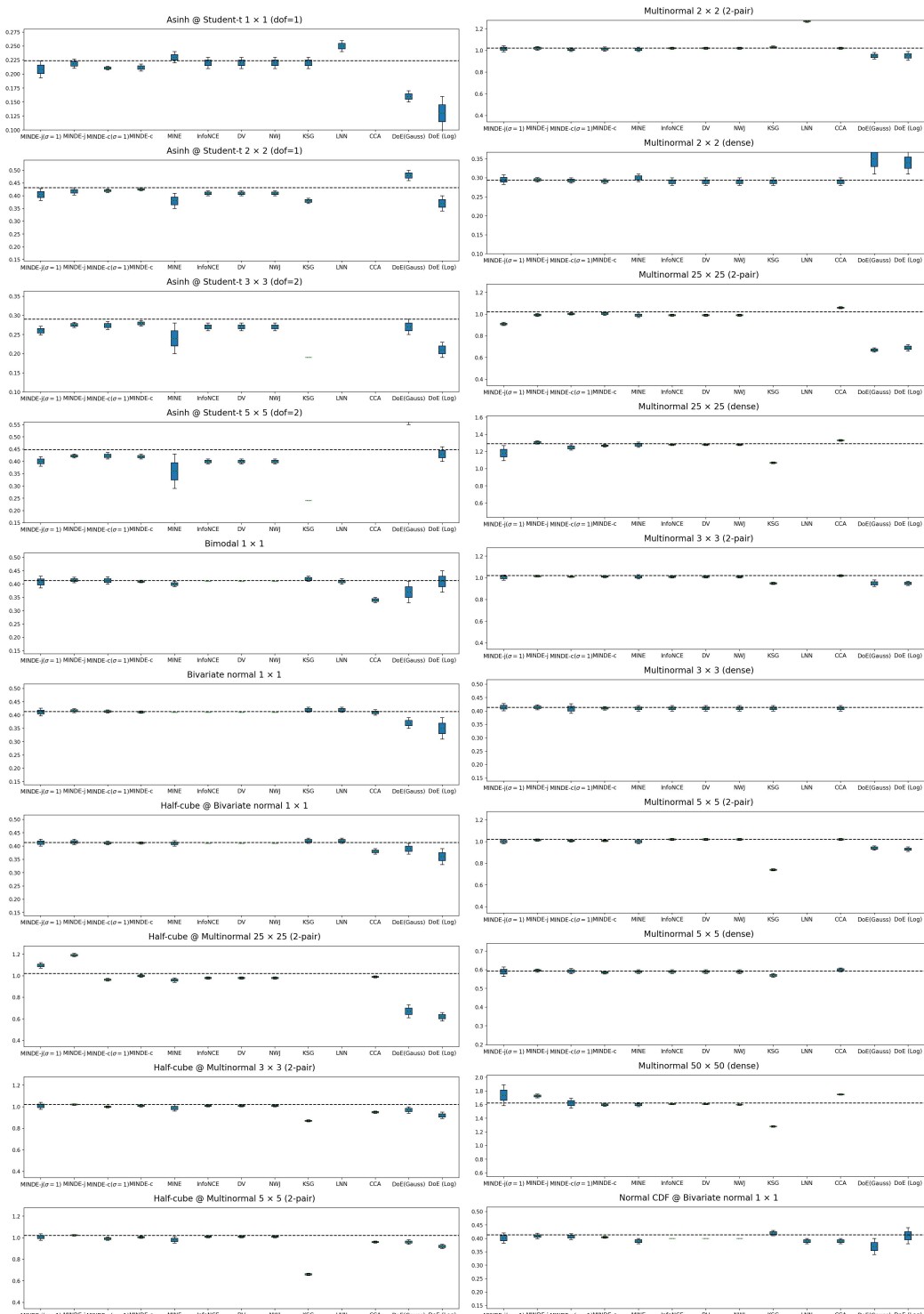

**Figure 3:** We report MI estimate results over 10 seeds for N =10000 for our method and competitors for training size 100k sample. A method absent from the depiction implies either non convergence during training or results out of scale

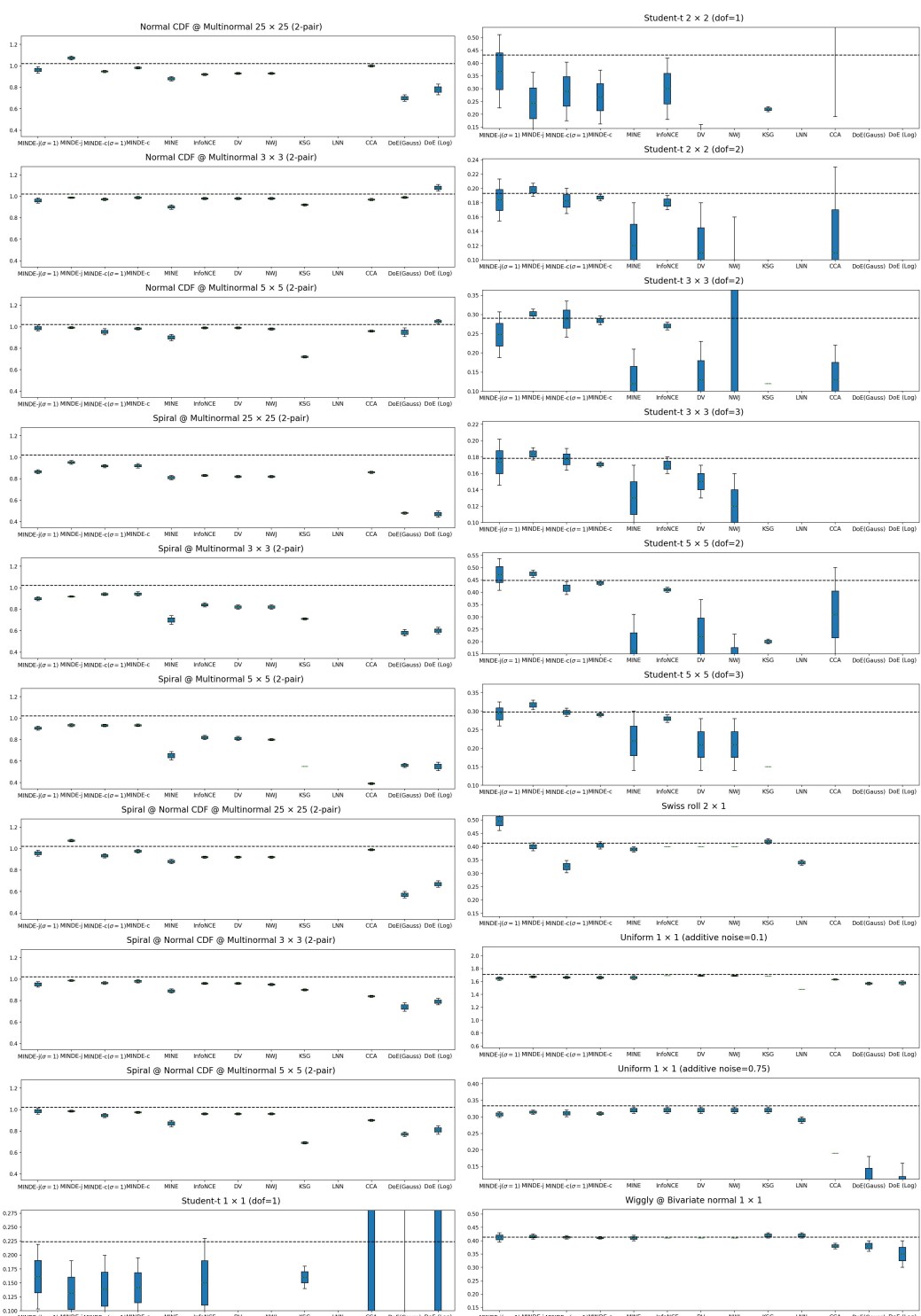

**Figure 4:** We report MI estimate results over 10 seeds for N =10000 for our method and competitors for training size 100k sample.

## D.3 TRAINING SIZE ABLATION STUDY

We here report, in Figures 5 to 8 the results of our ablation study on the training size, varying in the range 5k,10k,50k,100k.

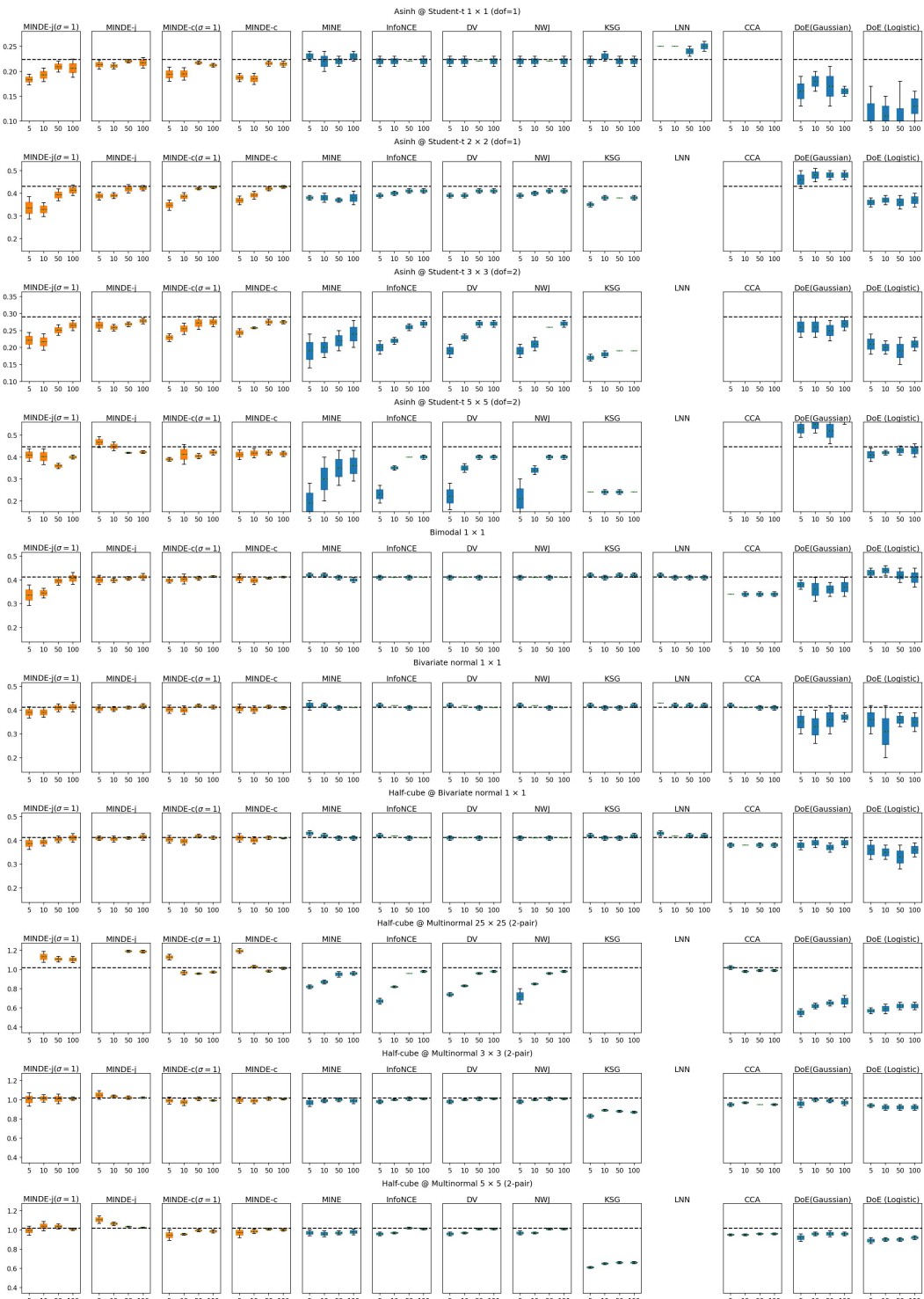

**Figure 5:** Training Size ablation study : We report MI estimate results for our method and competitors as a function of the training size used (5k,10k,50k,100k). For readability, we discard the baselines with estimation (error > 2 * GT) or high standard deviation. All results are averaged over 5 seeds. Due the benchmark size, we split the results into 4 figures each containing 10 benchmarks. A method absent from the depiction implies either non convergence during training or results out of scale. In this first plot we report tasks 1-10.

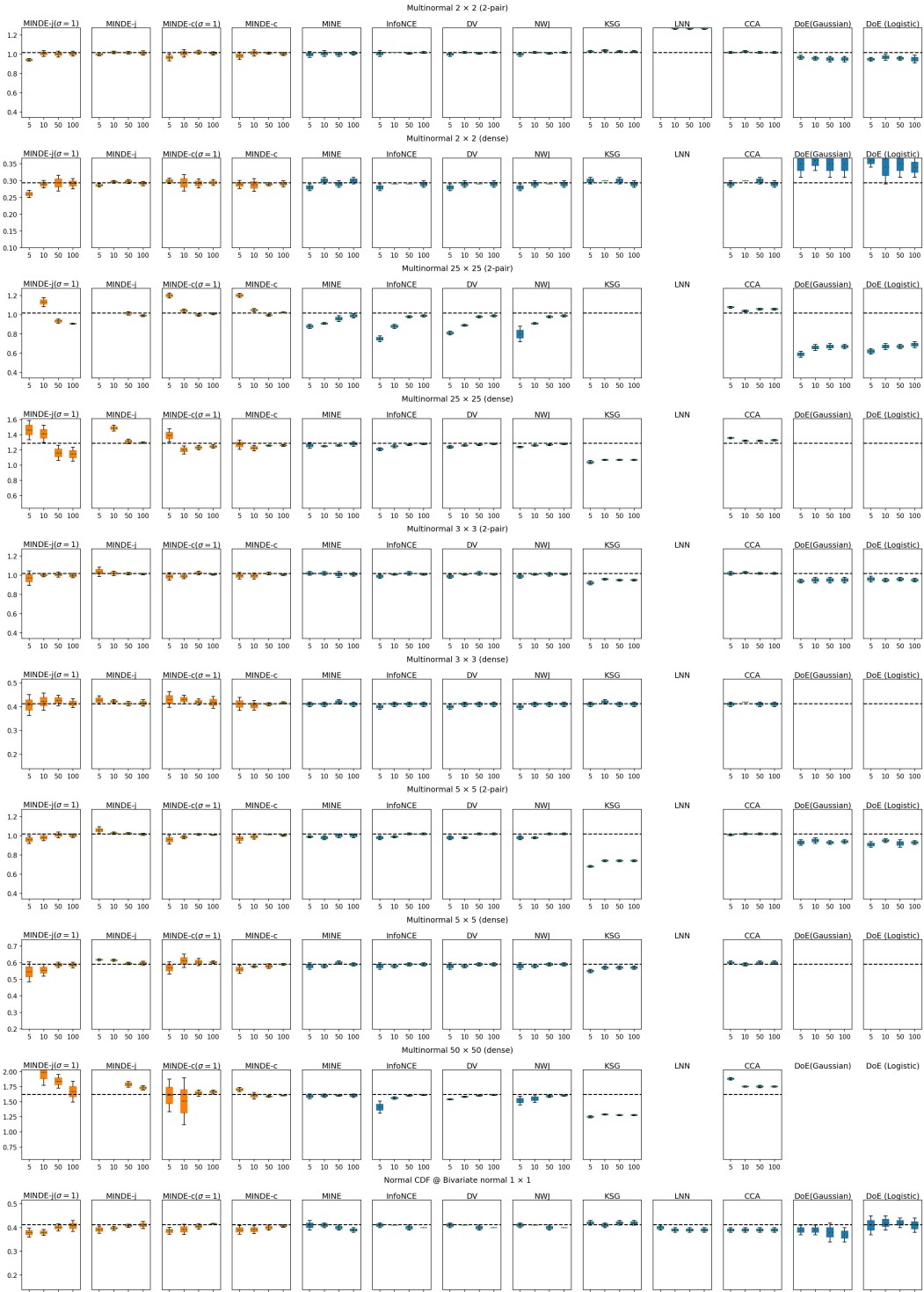

**Figure 6:** Part 2 of Figure 5, tasks 11-20.

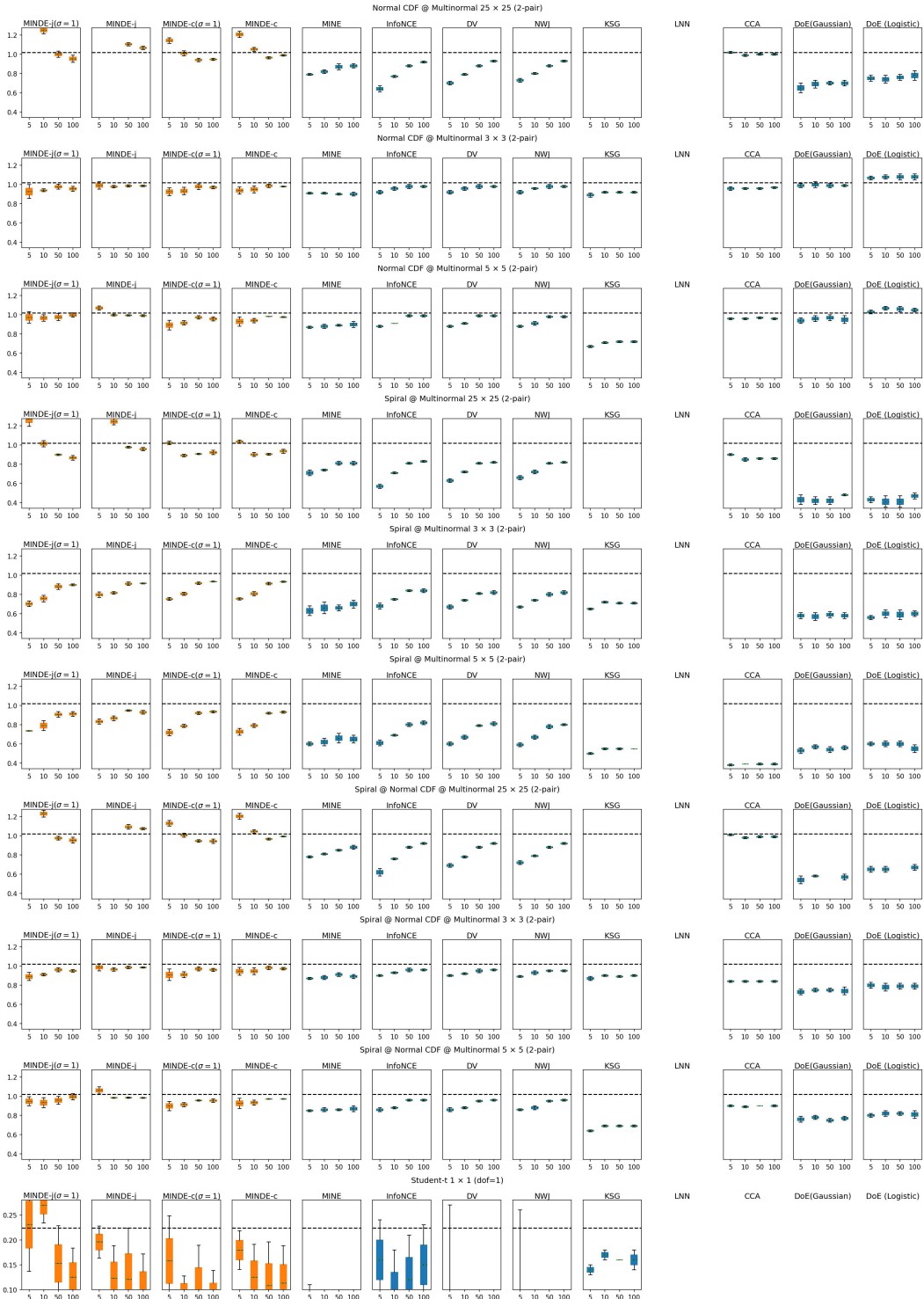

**Figure 7:** Part 3 of Figure 5, tasks 21-30.

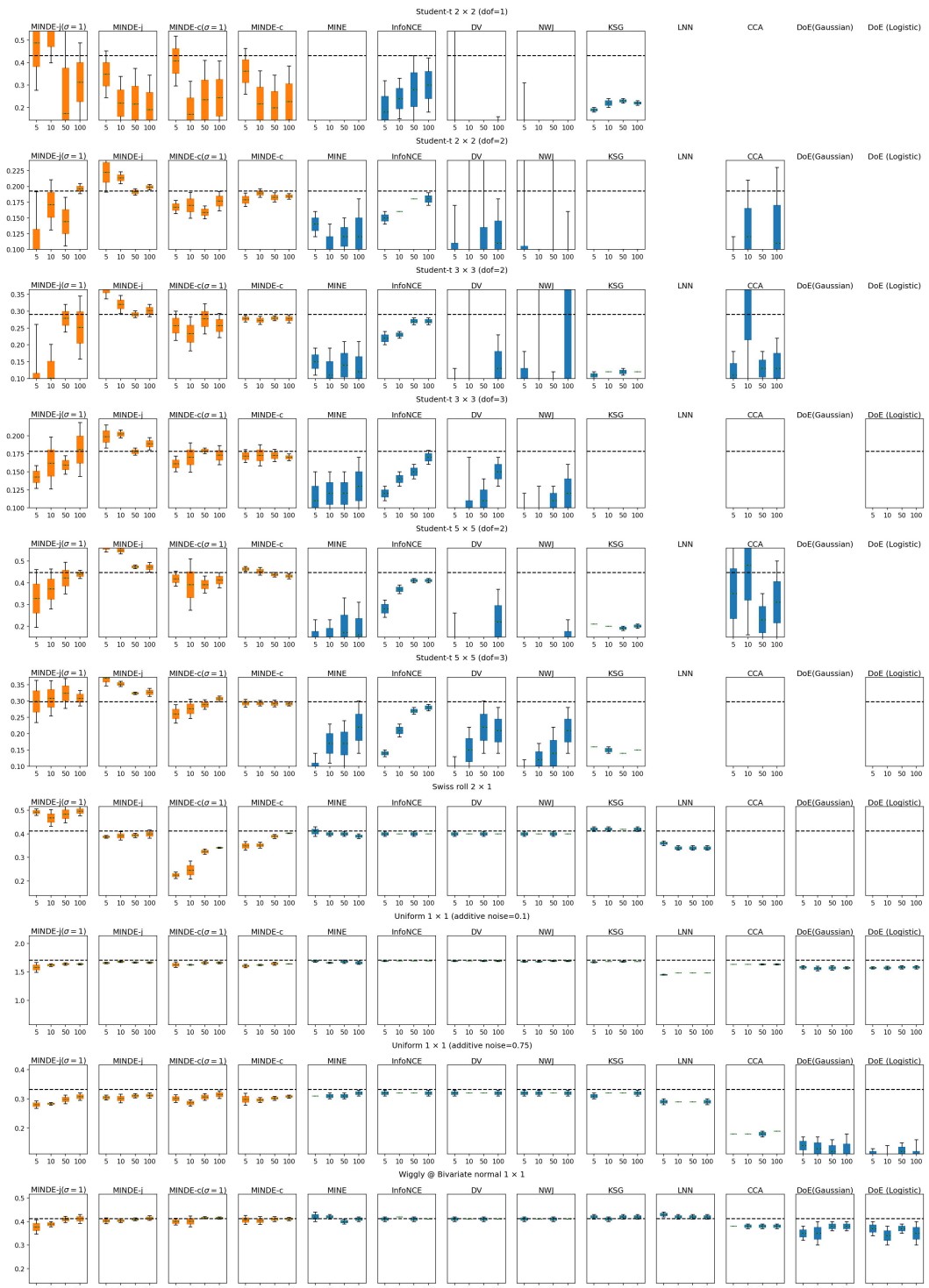

**Figure 8:** Part 4 of Figure 5, tasks 31-40.

## E   ANALYSIS OF CONDITIONAL DIFFUSION DYNAMICS USING MINDE

Diffusion models have achieved outstanding success in generating high-quality images, text, audio, and video across various domains. Recently, the generation of diverse and realistic data modalities

(images, videos, sound) from open-ended text prompts [Ramesh et al., 2022; Saharia et al., 2022; Rombach et al., 2022] has projected practitioners into a whole new paradigm for content creation. A remarkable property of our MINDE method is its generalization to any score based model. Then, our method can be considered as a plug and play tool to explore information theoretic properties of score-based diffusion models: in particular, in this section we use MINDE to estimate MI in order to explain the dynamics of image conditional generation, by analyzing the influence of the prompt on the image generation through time.

**Prompt influence of conditional sampling.** Generative diffusion models can be interpreted as *iterative* schemes in which starting from pure noise, at each iteration, refinements are applied until a sample from the data distribution is obtained. In recent work on **text conditional image generation** (image $A$, text prompt $B$) by Balaji et al. [2022], it has been observed that the role of the text prompt throughout the generative process has not constant importance . Indeed: *"At the early sampling stage, when the input data to the denoising network is closer to the random noise, the diffusion model mainly relies on the text prompt to guide the sampling process. As the generation continues, the model gradually shifts towards visual features to denoise images, mostly ignoring the input text prompt"* [Balaji et al., 2022]. Such claim has been motivated by carefully engineered metric analysis such as self and cross attention maps between images and text, as a function of the generation time, as well as visual inspection of the change in generated images when switching the prompt at different stages of the refinement.

Using MINDE, we can refine heuristic-based methods and produce a similar analysis using theoretically sound information theoretic quantities. In particular, we analyze the conditional mutual information $\mathrm{I}(A, B \mid X_\tau)$, being $X_\tau$ the result of the generation process at time $\tau$ (recall that the time runs backward from $T$ to $0$ during generation, and consequently $A = X_0$ and $B = Y_0$). Such metric quantifies, given an observation of the generation process at time $\tau$, how much information the prompt $B$ carries about the final generated image $A$. Clearly, when $\tau = T$, the initial sample is independent from both $A$ and $B$. Consequently, the conditional mutual information will coincide with $\mathrm{I}(A, B)$.

More formally, we consider the following quantity:

$$\mathrm{I}(A, B \mid X_\tau) = \mathrm{I}(A, B) - [\mathrm{I}(X_\tau, B) - \mathrm{I}(X_\tau, A \mid B)], \tag{26}$$

$$= \mathrm{I}(A, B) - [H(X_\tau) - H(X_\tau|B) - H(X_\tau|B) + H(X_\tau|A, B)], \tag{27}$$

$$= \mathrm{I}(A, B) - \mathrm{I}(X_\tau, B), \tag{28}$$

where Eq. (27) is simplified due to the Markov chain $A - X_0 - X_\tau$, so $H(X_\tau|A, B) = H(X_\tau|X_0, B) = H(X_\tau|B)$. Next, we use our MINDE estimator, whereby the marginal and conditional entropies can be estimated efficiently. The following approximation of the quantity in interest can be derived:

$$\mathrm{I}(A, B \mid X_\tau) \simeq \mathbb{E}_{\mathbb{P}^{\mu C}} \left[ \int_0^\tau \frac{g_t^2}{2} \left\| \tilde{s}_t^{\mu^A}(X_t) - \tilde{s}_t^{\mu^{A_{Y_0}}}(X_t) \right\|^2 \mathrm{d}t \right] \tag{29}$$

In our experiments, we also include a MINDE-($\sigma$) version which can be obtained similarly to Eq. (29).

**Experimental setting.** We perform our experimental analysis of the influence of a prompt on image generation using Stable Diffusion [Rombach et al., 2022], using the original code-base and pre-trained checkpoints.[2] The original Stable Diffusion model was trained using the DDPM framework [Ho et al., 2020] on images latent space. This framework is equivalent to the discrete-time version of VPSDE [Song et al., 2021]. Using the text prompt samples from LAION dataset Schuhmann et al. [2022], we synthetically generate image samples. We set guidance mechanism to 1.0 to ensure that

---

[2]https://huggingface.co/stabilityai/stable-diffusion-2-1

the images only contain text conditional content. We use 1000 samples and approximate the integral using a Simpson integrator [3] with a discretization over 1000 timesteps.

**Results.** We report in Figure 9 values of $I(A, B \mid X_\tau)$ as a function of (reverse) diffusion time, where $A$ is in the image domain and $B$ is in the text domain. In a similar vein to what observed by Balaji et al. [2022], our results indicate that $I(A, B \mid X_\tau)$ is very high when $\tau \simeq T$, which indicates that the text prompt has maximal influence during the early stage of image generation. This measurement is relatively stable at high MI values until $\tau \approx 0.8$. Then, the influence of the prompt gradually fades, as indicated by decreasing steadily MI values. This corroborates the idea that mutual information can be adopted as an exploratory tool for the analysis of complex, high dimensional distributions in real use cases.

The intuition pointed out by our MINDE estimator is further consolidated by the qualitative samples in Figure 10, where we perform the following experiment: we test whether switching from an original prompt to a different prompt during the backward diffusion semantically impacts the final generated images. We observe that changing the prompt before $\tau \simeq 0.8$ results almost surely with semantically coherent generated image with the second prompt. Instead, when $\tau < 0.8$, the second prompt influence diminishes gradually. We observe that for all the qualitative samples shown in Figure 10 the second prompt has no influence on the generated image after $\tau < 0.7$.

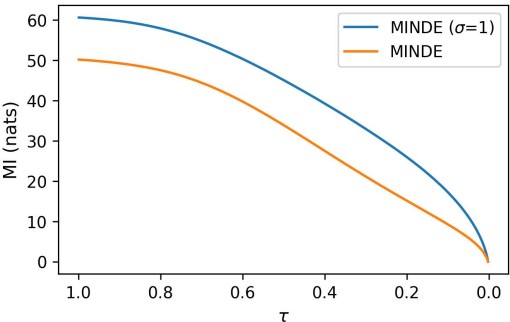

**Figure 9:** $I(A, B \mid X_\tau)$ as a function of $\tau$.

---

[3] https://docs.scipy.org/doc/scipy/reference/generated/scipy.integrate.simpson.html

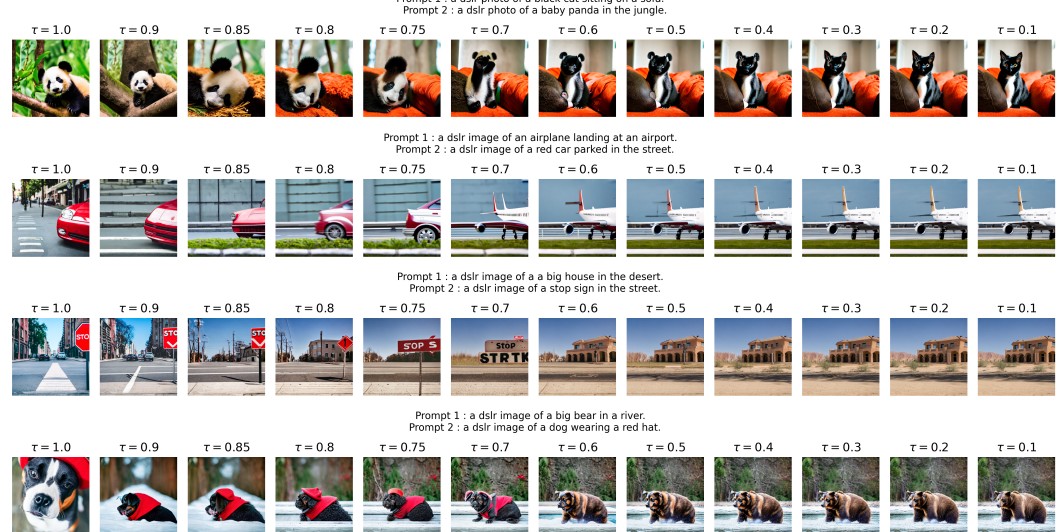

**Figure 10:** To validate the explanatory results obtained via the application of our MINDE estimator, we perform the following experiment: Conditional generation is carried out with *Prompt 1* until time $\tau$, whereas after the conditioning signal is switched to *Prompt 2*. We use the same Stable diffusion model as in the previous experiment with guidance scale set to 9.

## F   SCALABALITY OF MINDE

In this Section, we study the generalization of our MINDE estimator to more than two random variables. We consider the *information interaction* between three random variables $A$,$B$ and $C$, defined as:

$$\mathrm{I}(A, B, C) = \mathrm{I}(A, B) - \mathrm{I}(A, B|C) \tag{30}$$
$$= H(A) - H(A|B) - (H(A|B, C) - H(A, C)) \tag{31}$$

Estimation of such quantity is possible through a simple extension of Eq. (17) to three random variables, considering three parameters $\alpha, \beta, \gamma \in \{0, 1\}$.

In particular, we explore the case where the three random variables are distributed according to a multivariate Gaussian distribution: $A \sim \gamma_1$, $B = A + N_1$ (with $N_1 \sim \gamma_\epsilon$) and $C = A + N_2$ (with $N_2 \sim \gamma_\rho$). By changing the values of the parameters, it is possible to change the value of the interaction information. We report in Figure 11 the estimated values versus the corresponding ground truths, showing that MINDE variants can be effectively adapted for the task of information estimation between more than two random variables.

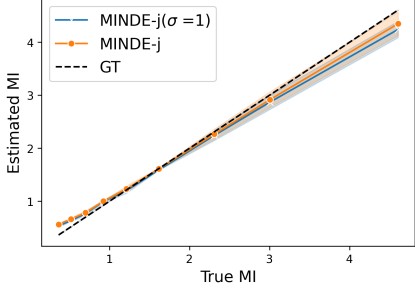

**Figure 11:** MI estimation results for MINDE-j on 3 variables

