# OpenReview forum: "MINDE: Mutual Information Neural Diffusion Estimation"
_ICLR.cc/2024/Conference — ICLR 2024 poster_

### Official Review · Reviewer_a1nb · 2023-10-21

**Soundness:** 4 excellent
**Presentation:** 3 good
**Contribution:** 3 good
**Rating:** 8
**Confidence:** 3

**Summary:**

The paper introduces Mutual Information Neural Diffusion Estimation as a family of mutual information estimation models based on estimating the difference of score functions.
The authors introduce and evaluate 4 variants based on the modeled scores (joins vs conditional) and the use of a standard normal as a reference for entropy computation.
An experimental section validates the theory on common mutual information estimation benchmarks, comparing MINDE against modern alternatives in literature and assessing self-consistency and compositionality.

**Strengths:**

1) The paper provides a solid, detailed derivation for MINDE rooted in SDE theory.

2) The experimental section effectively demonstrates the effectiveness of the proposed estimators on common benchmarks.

3) Although a Related Work section is missing, to the best of the reviewer’s knowledge, the paper includes references to all the relevant literature.

Overall I believe in the relevance and novelty of the submission and I am willing to increase my score whenever the authors address my main concerns.

**Weaknesses:**

# Main concerns
1) The experimental section benchmarks the estimators against common discriminative estimators such as MINE, NWJ, D-V, and InfoNCE which are designed as lower bounds of mutual information, but no comparison against generative estimators based on difference of entropies is provided [1,2]. Since MINDE, and in particular MINDE-$\sigma$, is based on the same principle, such a comparison seems natural.

2) The paper mentions previous similar work on diffusion-based mutual information estimation [3], which differs in the derivation and modeling choices. Nevertheless, this work is not included in the experimental comparison, and the advantages of MINDE are not further elaborated.

3) No discussion regarding the computational cost or challenges of training MINDE compared to the other models in the literature is included in the main text.

### Minor Remarks
1) The main text includes in-depth technical details with an extensive notation. If on one hand, this helps to verify the soundness of the derivation, on the other, it makes following the main derivation more difficult. I believe that submission could benefit by including additional intuition to guide the reader.

2) The plots in Figures 1 and 2 are quite small and difficult to read

### References
[1] McAllester, David, and Karl Stratos. "Formal limitations on the measurement of mutual information." International Conference on Artificial Intelligence and Statistics. PMLR, 2020.

[2] Song, Jiaming, and Stefano Ermon. "Understanding the limitations of variational mutual information estimators." ICLR, 2020.

[3] Kong, Xianghao, Rob Brekelmans, and Greg Ver Steeg. "Information-Theoretic Diffusion." ICLR, 2023.

**Questions:**

1) How does MINDE perform compared to classic generative estimators based on the difference of cross-entropies based on normalizing flows such as DoE in [1] and GM in [2]?

2) Can the author elaborate on the differences between MINDE and the work in [3]?

3) What are the main challenges when training the MINDE models? How does the training and inference cost (in terms of memory and computing) compare to the other neural estimators?

---

> ### Author Response · Authors · 2023-11-19
> **Rebuttal**
>
> We thank the reviewer for the positive feedback, which acknowledges soundness of the work and the relevance of experimental results. We submitted a new version of the manuscript, where the additional experiments and changes requested by the reviewer have been included.
>
> *Weaknesses: Main concerns
> 1.	The experimental section benchmarks the estimators against common discriminative estimators such as MINE, NWJ, D-V, and InfoNCE which are designed as lower bounds of mutual information, but no comparison against generative estimators based on difference of entropies is provided [1,2]. Since MINDE, and in particular MINDE-  is based on the same principle, such a comparison seems natural.*
>
> The connection between our generic estimator ( eq. (13) ) and the work of McAllester & Stratos [2020], is acknowledged in the submitted version of our paper (“This property, frees our estimation guarantees from the pessimistic results of McAllester & Stratos [2020]”). We do not consider this work as a competitor, but rather as a key literature resource which can help understanding, from a different perspective, one of the reasons why the estimator we propose has such good performance!
>
> Nevertheless, we do agree with the reviewer that an extra comparison with a generative approach using a DoE estimator could enrich our work. For this reason, we included a new direct comparison between MINDE variants and the official implementation of DoE (this appears both in the main paper, and in all Appendices). Even in this case, our method MINDE outperforms DoE, corroborating the idea that, while having an estimation which is neither an upper nor a lower bound has statistical advantages, a flexible model based on score functions is also key for the success of MINDE.
>
> *2.The paper mentions previous similar work on diffusion-based mutual information estimation [3], which differs in the derivation and modeling choices. Nevertheless, this work is not included in the experimental comparison, and the advantages of MINDE are not further elaborated.*
>
> As stated in our submitted version of the paper "a related estimator has recently appeared in the literature [Kong et al., 2022], although the technical derivation and objectives are different than ours".
>
> The work by Kong et al., which focuses mainly on likelihood estimation (with entropy estimation  being discussed as an interesting by-product), can be understood as a particular case of our generic framework.
>
> Indeed, our result: i) is immediately applicable to **any** forward SDE (whereas Kong et al. 2022  only focus on $ z_\gamma=\sqrt{\gamma}x+ \epsilon$ ) ii) is valid for generic KL divergences (we are not forced to estimate MI as difference of entropies, or difference outside, but also directly using the difference inside formulation) iii) allows to leverage SDE theory literature results about the convergence to 0 for large $T$ of the KL term between terminal distributions iv) allows the definition of a new method based on joint diffusion processes. While we believe that in principle the results of Kong et al. 2022 could be extended, to do so without explicitly introducing SDE theory would be cumbersome.  Summarizing, one way to look at the literature discussed in this answer is to consider the combination of the entropy estimator by Kong et al. 2022 and the technique of DoE to be an individual, particular instance of our proposed family of estimators.
>
> *3.No discussion regarding the computational cost or challenges of training MINDE compared to the other models in the literature is included in the main text.*
>
> Please see also a comment (and our answer) shared by Reviewer QLQ9. In summary, all neural based methods have comparable architectures, model sizes in terms of parameter count, and training times. Non parametric models, on the other hand, while being computationally efficient fail when distributions are complex, and high dimensional.
>
> It is very important to notice that our MINDE variants do not require the simulation of backward dynamics of a diffusion process, which is what is done in the literature for generative purposes.
>
> An additional note is in order. In the new results in Appendix F, we show an application use-case for our MINDE method, conditional version. In that case, training time is exactly 0, since we can use a pre-trained score model to estimate MI between real-world data (image and text prompts).

---

> > ### Author Response · Authors · 2023-11-19
> > **.**
> >
> > *Minor Remarks 1.	The main text includes in-depth technical details with an extensive notation. If on one hand, this helps to verify the soundness of the derivation, on the other, it makes following the main derivation more difficult. I believe that submission could benefit by including additional intuition to guide the reader.*
> >
> > This is a shared observation by reviewers. While our goal is being formal and technically sound, which is in contrast with providing a simplified description of our method, we did our best in our revised paper to help practitioners, by including additional pseudo-code with explanations (see Appendix D.1, D.2, Algorithms 1,2,3,4) , as well as providing the actual source code to reproduce the experiments. The source code will be made publicly available in later stages.
> >
> > *2.The plots in Figures 1 and 2 are quite small and difficult to read*
> >
> > Thanks for the editorial suggestion, we increased the font in the pictures in our revision!
> >
> > *Questions: How does MINDE perform compared to classic generative estimators based on the difference of cross-entropies based on normalizing flows such as DoE in [1] and GM in [2]? 2.	Can the author elaborate on the differences between MINDE and the work in [3]? 3.	What are the main challenges when training the MINDE models? How does the training and inference cost (in terms of memory and computing) compare to the other neural estimators?*
> >
> > Please see previous points for explicit and complete answers to your questions.

---

> > > ### Comment · Reviewer_a1nb · 2023-11-20
> > >
> > > I want to thank the authors for their comprehensive response, which satisfactorily addresses my primary concerns, and for the inclusion of additional experiments in their study. While I appreciate the updates, I note that the use of unimodal logistic and normal distributions in the DoE model might not adequately represent the complexity of the distributions in the benchmark. I suggest considering more flexible models, such as normalizing flows (specifically spline or NVP, as mentioned in [2]), to establish a stronger baseline for the DoE estimators.
> > > Despite this, I acknowledge the improvements made in the updated submission and I have increased my evaluation score accordingly.
> > >
> > > Additionally, I recommend that the author integrate their responses to questions 1 and 2 into the main text of the paper. This would provide future readers with immediate insights into these critical aspects of the research, further enhancing the paper’s role with respect to the existing literature.

---

> > > > ### Author Response · Authors · 2023-11-21
> > > > **Thank you for your feedback**
> > > >
> > > > Dear Reviewer a1nb,
> > > > thank you for your feedback. We will take your recommendation in to account in preparing a new version of the paper, both concerning additional experiments and editorial suggestions.
> > > >
> > > > Thanks a lot for acknowledging the value and improvements we made to our work and for raising the score accordingly.

---

### Official Review · Reviewer_QLQ9 · 2023-10-31

**Soundness:** 3 good
**Presentation:** 2 fair
**Contribution:** 2 fair
**Rating:** 6
**Confidence:** 2

**Summary:**

The paper proposes mutual information (MI) estimators based on diffusion models. This is achieved by representing MI as quantities that incorporate KL divergence which itself can be written as equations that involve score functions. Theoretical arguments justifying these claims are presented in the paper and experimental evaluation is performed on a recent benchmark dataset proposed for assessing MI estimators from multiple viewpoints (sparsity, dimensionality, long tails, and transformations). The results suggest that the developed estimator (MINDE) outperforms alternatives in most of these settings.

**Strengths:**

The problem considered is of critical importance in several applied and theoretical fields. Existing estimators either fail in high dimensions or require large amounts of data to provide precise estimates.

The results are quite impressive, the proposed estimator seems to outperform alternatives in most settings.

Several aspects of MI estimation that make the estimation challenging that were originally introduced in [1] such as sparsity, dimensionality, long tails, transformations, data processing, and consistency are considered in the experimental section making the empirical aspect of the paper strong.

[1] https://arxiv.org/abs/2306.11078

**Weaknesses:**

The organization of the paper makes it hard to follow. The measure theoretical notations make the paper inaccessible to the broader audience interested in using the estimator in applied settings.

The contributions are not fully clear. The connections between score, KL, MI, and H existed before. In addition, it's an established fact that diffusion process models are more powerful density estimators specifically in higher dimensions making it less surprising that the MI and H estimators are superior.

The comparisons lack several important aspects specifically in the context of MI estimation. The wall-clock runtime and the dataset size requirements are not particularly elaborated on in the paper.

**Questions:**

How does the method scale with the number of data points used for training? Can you make a plot of the test error as a function of the number of training data points used? The number of training data points can vary between 100, 1K, 10K, and 100K. Can you do this for a varying number of dimensions as well and report results for different estimators?

InfoNCE seems to be doing a very good job, can you come up with an overall score to rank the methods? Does InfoNCE also require a large dataset with the same size as yours?

Can you include time comparisons between different models? I assume learning the score functions from a diffusion process for the joint probability distribution would be an overkill if one only cares about the MI or H. What’s the training time comparison between different models and how do the authors account for computational resources used by various models?

Diffusion processes are known to outperform other density estimators in various settings. Therefore it’s no surprise if it achieves better an estimation of MI and H. That said, I’m having a hard time determining what the main contributions are. I imagine that the main contributions are representing MI and H as quantities that incorporate score functions. However, the connections between KL, score, MI, H are known results in the information theory literature [cite]. Is the extension of those results to the diffusion process (as opposed to generic densities) non-trivial? Is there something critical that I’m missing?

The paper would benefit from reorganization in my opinion. The notations used in the paper as well as the organization of the sections make it difficult to follow the arguments of the paper. The contributions start appearing very late in the paper and there is a large amount of background which might not be necessary for the main arguments. I suggest the following organization:
* Simple intro to diffusion models (no need to include the measure theoretical notation as it’s mainly used for the proofs and can be transported to the supplementary).
* Introducing the joint diffusion model (4.1) and MINDE.
* Making connections between MI and H, score functions, and KL divergence.

A recent paper [1] discusses that the absolute continuity assumption made in the paper might not hold given the architecture of the neural network used for approximating the score function (the cited paper discusses it in the context of functional variational inference but intuitively the same arguments should hold for diffusion process). In this case, the KL divergence will be infinity and the MI and H estimators developed will be ill-defined. Can the authors articulate the underlying assumptions further and explain what datasets can benefit from the estimators developed in the paper?

[1] https://arxiv.org/abs/2011.09421

---

> ### Author Response · Authors · 2023-11-19
> **Rebuttal**
>
> We thank the reviewer for appreciating our efforts in conducting an extensive experimental campaign. We hope that our careful rebuttal, as well as the new version of the manuscript will contribute in clarifying the reviewer's doubts.
>
> *Weaknesses:
> The organization of the paper makes it hard to follow. The measure theoretical notations make the paper inaccessible to the broader audience interested in using the estimator in applied settings.*
>
> The complexity of the measure theoretic notation is a shared concern, which we understand. However, our goal was to produce a rigorous, technical paper, and the theory which we build upon from the literature requires such rigor. Nevertheless, we do understand that making the paper more accessible to practitioners could greatly improve the impact of our work. For this reason, we prepared a new version of the paper including an extended section in Appendix D.1, D.2, where we include pseudo-code algorithms (Algorithms 1,2,3,4) for both training and estimation, and we included the anonymized source code, which will be made public in later stages.
> Finally, we also added new qualitative and quantitative results in Appendix F, showcasing a tangible application of MINDE to study complex, real-life datasets, in the context of the analysis of information dynamics of prompt-based generative models like Stable Diffusion.
>
> *The contributions are not fully clear. The connections between score, KL, MI, and H existed before. In addition, it's an established fact that diffusion process models are more powerful density estimators specifically in higher dimensions making it less surprising that the MI and H estimators are superior.*
>
> We are well aware that some of the “ingredients” of our work have been studied in the large body of work that exists on diffusion models, and we did our best to properly cite our sources. For the sake of producing a self-contained technical paper, we built upon existing, scattered connections and reunited them with a unique notation, to pave the way for our main contributions. Indeed, to the best of our knowledge, our MI estimator (and our variants) we propose in the paper was not known in the Machine Learning/ Statistics community before. Given the impressive empirical performance we obtained in our experimental campaign, we think our work is a useful contribution to the literature.
> We also add two new results in Appendix F and Appendix G. First, we showcase an application of MINDE by presenting new qualitative and quantitative results about the study of information dynamics of prompt-based generative models, which is a tangible application of MINDE to complex, real-life datasets (see Appendix F).
> Furthermore, we demonstrate that the theoretical contribution of our work, which allowed us to define the joint variant of MINDE, allows computing MI between more than two random variables, which is a property that to the best of our knowledge no other MI estimator shares. Our illustrative results use simple distributions for which computing the exact MI is possible (see Appendix G).
>
> *The comparisons lack several important aspects specifically in the context of MI estimation. The wall-clock runtime and the dataset size requirements are not particularly elaborated on in the paper.*
>
> Please, refer to answers to your questions below, where we elaborate on this comment.
>
> *Questions:
> How does the method scale with the number of data points used for training? Can you make a plot of the test error as a function of the number of training data points used? The number of training data points can vary between 100, 1K, 10K, and 100K. Can you do this for a varying number of dimensions as well and report results for different estimators? InfoNCE seems to be doing a very good job, can you come up with an overall score to rank the methods? Does InfoNCE also require a large dataset with the same size as yours?*
>
> Thank you for the suggestion. In the new version (see Appendix E.3 “Training size ablation study”) of the paper we include an extended experimental campaign varying the training size in the range 5k-10k-50k-100k. Overall, while InfoNCE performs well on larger training sizes, we observe that MINDE is robust to smaller training sizes when dealing with challenging distributions (see for example Figure 8 in Appendix E.3).

---

> > ### Author Response · Authors · 2023-11-19
> > **.**
> >
> > *Can you include time comparisons between different models? I assume learning the score functions from a diffusion process for the joint probability distribution would be an overkill if one only cares about the MI or H. What’s the training time comparison between different models and how do the authors account for computational resources used by various models?*
> >
> > There is an obvious difference in the complexity/estimation quality tradeoff between neural based estimators and classical estimators. Classical estimators have reduced computational cost but poor performance, especially for more complex datasets (see Appendices E.1, E.2 and E.3, and their Tables). Neural estimators, which include our MINDE variants, MINE, Info-NCE, etc…, have a computational complexity that is commensurate with the neural architectures used in their implementation. In our experimental campaign, we use neural estimators with models that have a comparable number of parameters (see Appendix D.3 and D.4). On our hardware (a single server equipped with A100 GPUs) we measured no significant difference in the training time among the various methods we tested.
> >
> > It is very important to notice that our MINDE variants do not require the simulation of backward dynamics of a diffusion process, which is what is done in the literature for generative purposes.
> >
> > An additional note is in order. In the new results in Appendix F, we show an application use-case for our MINDE method, conditional version. In that case, training time is exactly 0, since we can use a pre-trained score model to estimate MI between real-world data (image and text prompts).
> >
> > *Diffusion processes are known to outperform other density estimators in various settings. Therefore it’s no surprise if it achieves better an estimation of MI and H. That said, I’m having a hard time determining what the main contributions are. I imagine that the main contributions are representing MI and H as quantities that incorporate score functions. However, the connections between KL, score, MI, H are known results in the information theory literature [cite]. Is the extension of those results to the diffusion process (as opposed to generic densities) non-trivial? Is there something critical that I’m missing?*
> >
> > The connections between KL divergence, Entropy and MI are of course well-known concepts, and we believe our prose in the paper is clear about this. The reviewer is correct in noticing that our building blocks highlight an important connection to score functions. Our key contributions are the design of a family of estimators that rely on score functions (that are known to be well behaved), and that can either use a conditional formulation (which enables the practical use of pre-trained diffusion models, as done in Appendix F in our revised paper) or a novel joint formulation, which is extremely flexible, general and that has the potential to scale MI estimation to more than two random variables, a property that none of the existing work can claim (see additional results in the new Appendix G). Furthermore, we discuss the theoretical properties of our estimators and provide an extensive experimental campaign that include extremely recent and challenging benchmarks, self-consistency tests, and now also a real-life application example.
> >
> > *The paper would benefit from reorganization in my opinion. The notations used in the paper as well as the organization of the sections make it difficult to follow the arguments of the paper. The contributions start appearing very late in the paper and there is a large amount of background which might not be necessary for the main arguments. I suggest the following organization:
> > •	Simple intro to diffusion models (no need to include the measure theoretical notation as it’s mainly used for the proofs and can be transported to the supplementary).
> > •	Introducing the joint diffusion model (4.1) and MINDE.
> > •	Making connections between MI and H, score functions, and KL divergence.*
> >
> > While we do understand the reviewer’s comment and proposed presentation style, we politely disagree. This is in conflict with our goal of paving the way for a rigorous assessment of the properties of our estimators. We nevertheless aim at making our work more accessible to the general public. For this reason, we improved Appendix D.1, D.2, to clarify the practical implementation of our estimators, by providing the pseudo code (Algorithms 1,2,3,4) and related descriptions of the various algorithms we use for training, and for computing estimates. We also provide the source code of our work in the supplementary material, which will be made public in later stages.

---

> > > ### Author Response · Authors · 2023-11-19
> > > **.**
> > >
> > > *A recent paper [1] discusses that the absolute continuity assumption made in the paper might not hold given the architecture of the neural network used for approximating the score function (the cited paper discusses it in the context of functional variational inference but intuitively the same arguments should hold for diffusion process). In this case, the KL divergence will be infinity and the MI and H estimators developed will be ill-defined. Can the authors articulate the underlying assumptions further and explain what datasets can benefit from the estimators developed in the paper?*
> > >
> > > In the provided reference [1], the authors discuss how the KL divergence between Gaussian processes and parametric models can be infinite, which – to the best of our understanding –  is not directly applicable to our study.
> > > In our work, we study how to estimate the KL divergence between any two given measures. If such measures are not absolutely continuous, the divergence is indeed defined in our paper to be equal to infinity. In other words, our method works for datasets described by random variables whose joint and product of marginal measures are absolutely continuous.
> > >
> > > [1] https://arxiv.org/abs/2011.09421

---

> ### Comment · Reviewer_QLQ9 · 2023-11-21
>
> I thank the authors for replying to my comments revising the manuscript and performing the new experiments. The authors successfully addressed some of my concerns (scalability of the model in terms of its run time and its performance wrt the number of data points and dimensions).
>
> Given that the authors are aware of the existing theoretical connections used as the building blocks of their work, I suggest having a short paragraph or sentence and clearly mentioning what existed before and what's new in the paper. Although the references are included in various parts of the paper, the current manuscript might not precisely reflect the contributions made by the authors.
>
> I'll slightly increase my score because some of my concerns are not properly addressed. Specifically: (1) I still find the organization of the paper inaccessible. Although a new section in the appendix is added with the algorithmic description, this will not satisfactorily allow readers (even statisticians who are not familiar with stochastic processes and measure-theoretic notation) to follow the main ideas. (2) The results show that some of the existing estimators (such as Info-NCE) are on par with MINDE. They outperform MINDE in some datasets and provide lower variance estimates, while on other datasets MINDE does a better job. Given the new results presented in the appendix I find the authors' argument "while InfoNCE performs well on larger training sizes, we observe that MINDE is robust to smaller training sizes when dealing with challenging distributions" inaccurate.
>
> Regarding my last comment about absolute continuity, aren't the authors assuming that the neural net architecture is capable of estimating the density of the data distribution? Even if the true distributions of two variables $X$, $Y$ are absolutely continuous wrt one another, it might still be possible that the model does not capture that since the distributions that are achievable by the model given its underlying architecture might not be absolutely continuous wrt the data distributions. This is more of an intuitive comment rather than a precise mathematical argument, and given that the model is already working on a large number of datasets you can ignore this.

---

> > ### Author Response · Authors · 2023-11-22
> > **Thank you**
> >
> > We sincerely thank the reviewer for taking the time to read through our long rebuttal, and we are happy that some of the concerns have been successfully addressed. We will follow the suggestion to improve references to prior work and to emphasize our contributions.
> >
> > Again, thank you for your time and for the interaction with the authors.

---

### Official Review · Reviewer_hHbr · 2023-11-06

**Soundness:** 2 fair
**Presentation:** 3 good
**Contribution:** 2 fair
**Rating:** 6
**Confidence:** 2

**Summary:**

This work proposes *Mutual Information Neural Diffusion Estimation (MINDE)*, a novel method to estimate the mutual information (MI) between random variables. By first decomposing the KL divergence between two generic measures into two terms via its disintegration properties, then utilizing the Girsanov Theorem, MINDE provides a recipe that incorporates score-based diffusion models into the estimation of MI. The work provides four variants of MINDE, each based upon either conditional or joint diffusion processes, and presents experimental results that not only show the effectiveness in estimating MI accurately, especially on more challenging tasks (*e.g.*, spiral diffeomorphism), but also illustrate the robustness and reliability of the proposed method via self-consistency tests (including independency test, data-processing test, and additivity test) on the MNIST dataset.

**Strengths:**

1. The construction of the basic building blocks that establish the estimation of KL divergence and of the entropy is well organized and clearly written.
2. It’s interesting to see the SDE framework of diffusion models being used under the setting of MI estimation, which could inspire the research community to investigate diffusion models in new directions.

**Weaknesses:**

1. While the utilization of score-based diffusion models can be justified by the Girsanov Theorem, it’s unclear how they are used as **generative models** (*i.e.*, using the reverse-time SDE to generate samples) — it seems that only forward diffusion SDEs are needed, in order to train the score networks. Therefore, it’s a bit confusing when the authors wrote “we explore the problem of estimating MI using generative models” (Page 1), instead of something like “we explore the problem of estimating MI using score functions”.
2. Source code for the experiments is not provided.

**Questions:**

1. Could the authors discuss the connection between this work and MINE (Belghazi et al., 2018)? The title of this work seems to suggest a close connection with MINE, but the paper only provides the experimental results of MINE as one of the baseline models for MI estimation.
2. In Section 5.1, it is mentioned that using a larger training size shall “avoid confounding factors”. What do authors mean by “confounding factors”?
3. Might be a typo in Page 3: “Radon-Nikodyim derivative” shall be “Radon-Nikodym derivative”.

---

> ### Author Response · Authors · 2023-11-19
> **rebuttal**
>
> Thank you very much for your observations, and for noticing our original use of diffusion models for tasks beyond synthetic data generation.
>
> *Weaknesses:
> 1.	While the utilization of score-based diffusion models can be justified by the Girsanov Theorem, it’s unclear how they are used as generative models (i.e., using the reverse-time SDE to generate samples) — it seems that only forward diffusion SDEs are needed, in order to train the score networks. Therefore, it’s a bit confusing when the authors wrote “we explore the problem of estimating MI using generative models” (Page 1), instead of something like “we explore the problem of estimating MI using score functions”.
> We apologize for the confusion, as our sentence “we use generative models,  but with an original twist” might be misinterpreted indeed. Clearly, in our work we do not need the generative capabilities of diffusion models. The backward dynamics are however necessary to justify our central results, which allow estimating MI using score functions.*
>
> We modified the introduction, by referring to the distinction defined by the literature between discriminative and generative approaches (instead of models).
>
>
> *2.Source code for the experiments is not provided*
>
> We now include in the supplementary material an anonymized version of the source code. To further enhance reproducibility and clarity of the manuscript we also extended the Appendix D.1, D.2, with additional pseudo code (Algorithms 1,2,3,4) for the training and sampling procedures (see also concerns shared by Reviewer N2Vu).
>
> *Questions:
> 1.	Could the authors discuss the connection between this work and MINE (Belghazi et al., 2018)? The title of this work seems to suggest a close connection with MINE, but the paper only provides the experimental results of MINE as one of the baseline models for MI estimation*
>
> We consider MINE as an important contribution from the literature, and it indeed constitutes a strong baseline according to our experiments. There is however no direct connection between our method MINDE and that of MINE, it is just that the acronyms are similar, with the “D” indicating the use of “diffusion processes”.
>
>
> *2.	In Section 5.1, it is mentioned that using a larger training size shall “avoid confounding factors”. What do authors mean by “confounding factors”?*
>
> Not all the methods are equally resilient to smaller training sizes. In particular, the competitors we consider in our experiments perform worse than our MINDE variants for smaller (5k,10k,50k) training sizes (see also comments by  Reviewer QLQ9). In our submission, our goal was to perform a fair comparative analysis, and we chose the large training size regime to boost the performance of our competitors. We now present a new set of experimental results (see Appendix E.3 “Training size ablation study”), where we show that MINDE variants outperform competitors in a wide range of training sizes.
>
>
> *3.Might be a typo in Page 3: “Radon-Nikodyim derivative” shall be “Radon-Nikodym derivative”.*
>
> Thanks for spotting the typo! We fixed it.

---

> > ### Comment · Reviewer_hHbr · 2023-11-21
> > **Thank You Authors**
> >
> > Thank you for your responses to my review $-$ my questions have been sufficiently addressed. I will keep my current score as the rating is already positive.

---

> > > ### Author Response · Authors · 2023-11-21
> > > **Thank you for the feedback!**
> > >
> > > We thank the reviewer for acknowledging our rebuttal, and we are happy to learn our answers were satisfactory. Would you be so kind to consider also the numerous extra experiments we performed to further improve our submission which include:
> > > * a new section (Appendix F) on a practical use case for MINDE, where we showcase how it can be used to study prompt-based image generation
> > > * a new section (Appendix G) that illustrates how our joint diffusion process variant enables scalability in the number of random variables
> > >
> > > We believe that these, and the other numerous improvements we implemented following all reviewers comments, have really improved our work!
> > > Despite the score leaning on the positive side, and we thank you for this, we hope that our improved version of the paper could provide strong evidence of its merits.

---

### Official Review · Reviewer_N2Vu · 2023-11-06

**Soundness:** 4 excellent
**Presentation:** 2 fair
**Contribution:** 3 good
**Rating:** 6
**Confidence:** 3

**Summary:**

The authors derive an estimator of mutual information using neural diffusion.

**Strengths:**

I really like that the authors used the Czyz benchmark data, and also the consistency tests.
I also appreciate the creativity of the theoretical advancement, though I don't understand it (see below).

**Weaknesses:**

I was super excited to read this paper, because I love thinking about mutual information and entropy, and have recently been working on some related issues.   The ideas are intriguing, and the results are impressive. So, the rest of this review will focus on the issues for me understanding the methods and results.

1. The biggest issue for me is that I almost immediately got lost.  I know information theory pretty well, I learned it from Fred Jelinek before he died. That said, I know very little about diffusion processes and SDEs. My main confusion with this paper was about connecting the math on diffusion processes to the process of estimating MI. There is a leap, which I am willing to believe is justified, that I completely missed.  Why are we talking about a filtration and an Ito process at all?  How do they related to the joint distribution F_{X,Y}? I read the words in Section 2-4, but was completely lost.  To be fair, I was also lost the first time I read the KSG paper (https://journals.aps.org/pre/abstract/10.1103/PhysRevE.69.066138).  I imagine lots of people might have followed the logic and derivation completely. But not me, I just didn't get it. And I spent some time trying to figure it out, as I'd like to get it, it seems cool, and within the realm of possibilities that I did get it, but I didn't. I thought maybe reading Appendix D would help me, but it didn't really help either.  In the end, I don't quite know what you did, or why you did it.  I would love something like Algorithm 1 and 2, which perhaps points to subroutines for how each relevant quantity is computed. For example, I don't see how to do "r.h.s. Eq. (16)". Where does 'g' come from, or k, or T? etc.

2. In terms of the numerical results, I think I understand them, which was exciting for me! Figure 1 shows that MINDE works about as well as other things on relatively easy problems where there is enough data, and slightly better on a spiral dataset when MI is high.  That's cool as far as it goes.  I'm always interested in *finite sample properties* for my estimators, because I always have finite data.  In particular, I often work on biomedical problems, in which sample sizes are typically hundreds.  So, I would be much more interested in seeing plots showing accuracy as a function of sample size, especially for the "easy" ones where many different estimators are getting the right answer. This introduces additional information about convergence rates. The fact that 2 high-dimensional simulations showed it does as well as other things, and one showed it is slightly better for some parametrization, I found not that compelling.

3. I understand why the authors say that the benchmark consists of 40 tasks.  However, in my opinion, this wording is confusing and misrepresenting the work. I would say that there are about 10 different tasks, with an average of 4 different parameterizations per task. Consider, for example, our paper https://elifesciences.org/articles/41690. We describe 20 tasks, but in Figure 2, we should many different parameterizations (dimensions) per task. Claiming that we had more than 20 tasks, in my opinion, would not be in integrity.  Varying parameters, dimensions, and sample sizes for a particular task is important, but claiming that each different parameterization is a different task seems inappropriate to me. Of note, the Czyz et al paper from which the tasks are extracted never seems to make such a claim. Rather, they name about 10 different tasks explicitly.

4. The claim that MINDE outperforms other approaches on 35/40 tasks I also question. No errorbars are provided.  While this is typical in machine learning benchmark comparisons, I think the practice is ill-advised and misleading.  Without errorbars, there is no evidence that if one ran the exact same code again, how likely is it that the results would be similarly ordered.  There is a rich history of non-parametric tests for evaluating whether one estimator tends to be better than another, and I would encourage the authors to only claim something is better when there is statistical evidence supporting the claim.

5. Directly using kernel density estimators and plugging them into the MI equations, or using the standard approaches to estimating mutual information (eg, KSG, which is included in sklearn), to compare with the neural methods, would also be important.

**Questions:**

Things that would inspire me to increase the score:
1. Explain (more/better) why they are talking about SDEs for estimating MI.
2. Perform statistical analyses and/or errorbars indicating whether MINDE is significantly better than anything else on any particular simulation.
3. Clarify how many different settings were considered, and discuss from that perspective.
4. More clear/complete pseudocode.

---

> ### Author Response · Authors · 2023-11-19
> **rebuttal**
>
> We thank the reviewer for acknowledging the originality of our method, and the high relevance of the experimental results. We hope our rebuttal, along with the revised version of our paper, will clarify all doubts and questions of the reviewer.
>
> *Weaknesses:
> I was super excited to read this paper, because I love thinking about mutual information and entropy, and have recently been working on some related issues. The ideas are intriguing, and the results are impressive. So, the rest of this review will focus on the issues for me understanding the methods and results.
> 1.	The biggest issue for me is that I almost immediately got lost. I know information theory pretty well, I learned it from Fred Jelinek before he died. That said, I know very little about diffusion processes and SDEs. My main confusion with this paper was about connecting the math on diffusion processes to the process of estimating MI. There is a leap, which I am willing to believe is justified, that I completely missed. Why are we talking about a filtration and an Ito process at all? How do they related to the joint distribution F_{X,Y}? I read the words in Section 2-4, but was completely lost. To be fair, I was also lost the first time I read the KSG paper (https://journals.aps.org/pre/abstract/10.1103/PhysRevE.69.066138). I imagine lots of people might have followed the logic and derivation completely. But not me, I just didn't get it. And I spent some time trying to figure it out, as I'd like to get it, it seems cool, and within the realm of possibilities that I did get it, but I didn't. I thought maybe reading Appendix D would help me, but it didn't really help either. In the end, I don't quite know what you did, or why you did it. I would love something like Algorithm 1 and 2, which perhaps points to subroutines for how each relevant quantity is computed. For example, I don't see how to do "r.h.s. Eq. (16)". Where does 'g' come from, or k, or T? etc.*
>
> We apologize with the reviewer for the dense and rather involved notation of our paper, it is indeed not an easy paper to digest! The technical introduction of measure theoretic quantities serves the purpose of being able to rigorously assess validity of the central claims of our work (equation (10) and the estimator equation (11) ). Indeed, central to the technical derivation of the estimator, are path measures of diffusion processes, their time reversal, and the *ratio* of such paths (expressed thanks to Girsanov Theorem). All these results, which are built on a growing body of work from the literature, require the rigorous definitions used in our paper. We also made a particular effort in harmonizing and unifying the notation, which is intended to simplify life for the reader, instead of asking readers to dive into several other articles from the literature, each with their own notation.
> Our work sets the basis to estimate the KL divergence between ANY pair of measures, and we focus in particular on the case of Mutual Information. This can be cast as the KL between joint and marginals, or the KL between conditional and marginals (with an extra expected value). All the different formulations correspond to variants of MINDE that we experimentally validate.
> We do understand that our stylistic choice, while necessary for a technically sound paper, can harm accessibility of our work to a broader audience. For this reason, as correctly hinted by the reviewer, we included in the new version of the manuscript (see Appendix D.1, D.2, Algorithms 1,2,3,4) an extended version of the pseudocode for our methods where i) we clarify explicitly how to use “r.h.s of (16)” and ii) we accurately describe the training subroutines, referring to the equations in the main paper.
> Finally, concerning  “Where does $g$ come from, or $k$, or $T$? etc.”, these are parameters which have been introduced throughout the manuscript. In particular $g_t$ is the diffusion coefficient (eq. (1)), $k_t$ is defined in the paragraph after eq. (14) (at the end of “ Now, the score function…”) and $T$ is the duration of the diffusion process, introduced at the very beginning before eq. (1). In practical experiments, we adopt one of the standard SDE typically adopted in the literature (VP-SDE, [Song2021]) for which the aforementioned quantities have very simple expressions.

---

> > ### Author Response · Authors · 2023-11-19
> > **.**
> >
> > *In terms of the numerical results, I think I understand them, which was exciting for me! Figure 1 shows that MINDE works about as well as other things on relatively easy problems where there is enough data, and slightly better on a spiral dataset when MI is high. That's cool as far as it goes. I'm always interested in finite sample properties for my estimators, because I always have finite data. In particular, I often work on biomedical problems, in which sample sizes are typically hundreds. So, I would be much more interested in seeing plots showing accuracy as a function of sample size, especially for the "easy" ones where many different estimators are getting the right answer. This introduces additional information about convergence rates. The fact that 2 high-dimensional simulations showed it does as well as other things, and one showed it is slightly better for some parametrization, I found not that compelling.*
> >
> > First, let us clarify that the experimental validation proposed in the benchmark [Czyz et al. 2023 ] is not limited to two dimensional distributions. Indeed the considered dimensionality varies for the different tasks, reaching a joint dimension of 100.
> > We do agree with the reviewer that the limit of the large data regime allows only partial assessment of the performance of different estimators. In the submitted version of the paper we considered the setting of 100k training samples (see also a comment on this shared by Reviewer QLQ9). Indeed, for smaller training sizes competitors were less stable than our method; then, for the sake of a fair comparison by limiting stability problems,  we selected a large training size. Thanks to the reviewer’s comment, we now include in the new version of the paper (see Appendix E.3 “Training size ablation study”) results for a large range of training sizes, i.e., 5k, 10k, 50k and 100k. From these new results, see Figures 5,6,7,8, it is evident how (as expected) all methods perform worse when training size is smaller; nevertheless, MINDE variants are generally more robust than other competitors.
> >
> >
> > *I understand why the authors say that the benchmark consists of 40 tasks. However, in my opinion, this wording is confusing and misrepresenting the work. I would say that there are about 10 different tasks, with an average of 4 different parameterizations per task. Consider, for example, our paper https://elifesciences.org/articles/41690. We describe 20 tasks, but in Figure 2, we should many different parameterizations (dimensions) per task. Claiming that we had more than 20 tasks, in my opinion, would not be in integrity. Varying parameters, dimensions, and sample sizes for a particular task is important, but claiming that each different parameterization is a different task seems inappropriate to me. Of note, the Czyz et al paper from which the tasks are extracted never seems to make such a claim. Rather, they name about 10 different tasks explicitly.*
> >
> > We apologize for the confusion. We do agree with the reviewer that the wording could be clearer, and we updated the paper accordingly.
> >
> >
> > *The claim that MINDE outperforms other approaches on 35/40 tasks I also question. No errorbars are provided. While this is typical in machine learning benchmark comparisons, I think the practice is ill-advised and misleading. Without errorbars, there is no evidence that if one ran the exact same code again, how likely is it that the results would be similarly ordered. There is a rich history of non-parametric tests for evaluating whether one estimator tends to be better than another, and I would encourage the authors to only claim something is better when there is statistical evidence supporting the claim.*
> >
> > We now have included in Appendix E.2 and E.3, for completeness, the results concerning the standard deviations of all our experiments. Summarizing, the new numerical evidence suggests that indeed MINDE variants have a statistically significant better performance than competitors.
> >
> > *Directly using kernel density estimators and plugging them into the MI equations, or using the standard approaches to estimating mutual information (eg, KSG, which is included in sklearn), to compare with the neural methods, would also be important.*
> >
> > We completely agree with the reviewer! These results are already present in the submitted version of our paper (Appendix E.1). Unsurprisingly, classical methods perform worse than the more recent neural approaches. Also in this case our claims are now statistically grounded by observing at the standard deviations of the results (which were not present in the submitted version of the paper).

---

> > > ### Author Response · Authors · 2023-11-19
> > > **.**
> > >
> > > *Questions:
> > > Things that would inspire me to increase the score:
> > > 1.	Explain (more/better) why they are talking about SDEs for estimating MI.
> > > 2.	Perform statistical analyses and/or errorbars indicating whether MINDE is significantly better than anything else on any particular simulation.
> > > 3.	Clarify how many different settings were considered, and discuss from that perspective.
> > > 4.	More clear/complete pseudocode.*
> > >
> > > Thank you very much for the summary, which we followed point by point. Note also that we provide new qualitative and quantitative results about the study of information dynamics of prompt-based generative models, showcasing a tangible, practical application of MINDE to study complex, real-life datasets.

---

> > > > ### Author Response · Authors · 2023-11-22
> > > > **Feedback**
> > > >
> > > > Dear Reviewer N2Vu,
> > > >
> > > > We prepared a thorough rebuttal and addressed all your questions, including many additional experiments.
> > > >
> > > > We hope you will have time to go through our answers, and that these will indeed inspire you to increase our score. We would really appreciate your feedback!
> > > >
> > > > Thank you very much for your work!

---

> > > > ### Comment · Reviewer_N2Vu · 2023-11-22
> > > > **Addressed some concerns**
> > > >
> > > > 1. Is not yet satisfactory for me, I desire further re-writes
> > > > 2. Analysis is performed, I desire more effective summaries
> > > > 3. Satisfied
> > > > 4. Better, though the equations in the pseudocode were confusing to me, as I did not understand where they all came from. What would be great would be if all equations that showed up in the pseudocode showed up in the main text with explanations, and then the pseudocode simply pointed to it.  For example, eq. 16 shows up in the main text, and something else shows up in the pseudocode, without an explanation of how to get from eq. 16 to that.
> > > >
> > > > I will revise your scores accordingly.  That said, I hope that you significantly revise the text, and get another critical pair of eyes on it, because I believe the impact of your work is currently limited by the writing style.

---

> > > > > ### Author Response · Authors · 2023-11-23
> > > > > **Thank you!**
> > > > >
> > > > > Dear Reviewer N2Vu,
> > > > > we would like to thank you for the energy and effort you have put in evaluating our work.
> > > > > We will do our best to follow your suggestions when preparing a possible camera ready version of this paper.
> > > > >
> > > > > Thank you again, the Authors.

---

> > > ### Comment · Reviewer_N2Vu · 2023-11-22
> > > **Numerical results are clearer**
> > >
> > > I commend the authors for improving the clarity of the numerical results. However, there are now *many* new figures, each with *many* columns, and with very small fonts, and using boxplots.  It is too much for me to parse easily.  Things I want instead:
> > >
> > > 1. I want a main figure showing accuracy vs sample size that summarizes all that stuff.
> > > 2. Never show boxplots, they obscure the details, if there are 10 simulations, simply show jittered scatterplots or beeswarm plots.
> > > 3. Make all text in figures no less than 1-2 points smaller than the main body fonts, including legends, tick labels, etc.
> > > 4. Use standard methods to show errorbars in tables, eg, show them in parentheses.
> > > 5. Only color things that are significantly different.

---

> > ### Comment · Reviewer_N2Vu · 2023-11-22
> > **Thank you**
> >
> > It is clearer (to me) now, the appendix revisions helped.  However, it is still not clear.  I invite you to inquire about the following comment:
> >
> > "We do understand that our stylistic choice, while necessary for a technically sound paper, can harm accessibility of our work to a broader audience. "
> >
> > Is your style choice truly "necessary" for a technically sound paper?  I do not believe that is the case.  I believe the stylistic choice you have made renders the paper inaccessible for a large subset of the intended community.
> >
> > Further changes I would recommend:
> >
> > 1. Section 2 comes out of nowhere for me, it is only justified to look at diffusion processes after we understand that you are using them in Section 3, so I'd re-order the contents.
> > 2. All terminology be define prior to its first use.  eg, you introduce \mu^A as a generic measure, and then start writing about \mathbb{P}^{\mu^A}, without defining it first (or maybe ever). Similarly for "#", and the hat notation.
> > 3. Restrict yourselves to 1 superscript, not 2.
> >
> > If you made all those changes, maybe I'd be able to understand.  Please note that I really *want* to understand, and I've invested a lot of energy in understanding, and still failed.  I firmly believe you are capable of writing in a way that I would understand, and want you to, both for my sake, and yours.

---

> ### Public Comment · ~Xianghao_Kong1 · 2024-07-31
> **Bridge the gap between Info and Diffusion**
>
> Hi there,
>
> As you mentioned in Weakness 1, I would like to recommend the ICLR 2023 paper "[Information-Theoretic Diffusion](https://arxiv.org/abs/2302.03792)". This paper presents a novel diffusion model constructed from an information theory perspective. It introduces a new mathematical foundation to bridge the gap between diffusion and information theory.
>
> In essence, the backbone (UNet) in diffusion models acts as a noise estimator or denoiser (MMSE). The connection between Mutual Information (MI) and MMSE creates a perfect match for both concepts.
>
> For more details and explanations, please visit our [GitHub Repo](https://github.com/kxh001/ITdiffusion) and [simplified demo](https://github.com/gregversteeg/InfoDiffusionSimple).

---

### Author Response · Authors · 2023-11-19
**Message to the reviewers**

Dear Reviewers,

We sincerely appreciate the time and effort you have dedicated to reviewing our paper. Your insights and constructive questions have been invaluable in guiding our revision. In this rebuttal, we aim to address all concerns comprehensively, while underscoring the strengths of our work, the necessity of our chosen notation for scientific rigor, and the breadth of our experimental validation.

A common question among the reviews concerns the rather involved technical notation we used to describe our method. Indeed measure-theoretic language, while crucial for the scientific exactness and theoretical foundation of our work, may pose challenges in terms of accessibility to a broader public. This complexity is not gratuitous but is important to establish the validity of our central claims, especially concerning the estimator equations, its properties and their derivations using path measures of diffusion processes. In response to the feedback, we have made efforts to enhance the clarity. This includes an expanded section in the Appendix including additional pseudocode, which lays out the application of our theoretical constructs in a more practical format, and the release of the associated source code for our work. We hope that these revisions will make our work more accessible to the practitioners community, without compromising the rigor of its derivation.

Another focal point of the rebuttal is an extensive set of new experiments which we included in our revised paper. These additional experiments highlight the effectiveness and versatility of our method MINDE for practical endeavors. First, we've broadened the scope of our experimental validation to include various training sizes (5k,10k,50k), providing a comprehensive picture of MINDE's complexity/performance relative to existing methods. Moreover, we also include additional experiments using an instance of the generative DoE method, as suggested by one reviewer. The new results indicate that our method also outperforms competitors in the realm of generative approaches.

We also present new qualitative and quantitative results about the study of information dynamics of prompt-based generative models, showcasing a tangible application of MINDE to study complex, real-life datasets.

Furthermore, we demonstrate that the theoretical contribution of our work, which allowed us to define a MINDE variant based on joint diffusion processes, allows computing MI between more than two random variables, which is a property that to the best of our knowledge no other MI estimator shares.

---

### Comment · Area_Chair_KFLn · 2023-11-22

Dear all,

The author-reviewer discussion period is about to end.

@authors: If not done already, please respond to the comments or questions reviewers may further have. Remain short and to the point.

@reviewers: Please read the author's responses and ask any further questions you may have. To facilitate the decision by the end of the process, please also acknowledge that you have read the responses and indicate whether you want to update your evaluation.

You can update your evaluation positively (if you are satisfied with the responses) or negatively (if you are not satisfied with the responses or share other reviewers' concerns). Please note that major changes are a reason for rejection.

You can also keep your evaluation unchanged. In this case, please indicate that you have read the responses, that you do not have any further comments and that you keep your evaluation unchanged.

Best regards,
The AC

---

### Meta-Review · Area_Chair_KFLn · 2023-12-09

**Metareview:**

The reviewers unanimously recommend acceptance (8-6-6-6). The paper proposes a novel approach to estimate the mutual information between two random variables, based on an original interpretation of the Girsanov theorem. The reviewers note the creativity of the theoretical results, the quality of the empirical evaluation and the improvements over previous baselines. The author-reviewer discussion has been constructive and has led to a number of improvements to the paper. The only remaining concern is about the technicality of the presentation, which could be improved to make the paper more accessible to a broader audience. We encourage the authors to find the right balance between technicality and accessibility in the final version of the paper.

**Justification For Why Not Higher Score:**

Some concerns remain regarding the presentation, which is rigorously correct, but hard to follow.

**Justification For Why Not Lower Score:**

The reviewers unanimously recommend acceptance.

---

### Decision · Program_Chairs · 2024-01-16

Accept (poster)